# Profiling of the *Helicobacter pylori* redox switch HP1021 regulon using a multi-omics approach

Mateusz Noszka [1], Agnieszka Strzałka [2], Jakub Muraszko [1], Rafał Kolenda [3,4], Chen Meng [5], Christina Ludwig [5], Kerstin Stingl [6] & Anna Zawilak-Pawlik [1] ✉

The gastric human pathogen *Helicobacter pylori* has developed mechanisms to combat stress factors, including reactive oxygen species (ROS). Here, we present a comprehensive study on the redox switch protein HP1021 regulon combining transcriptomic, proteomic and DNA-protein interactions analyses. Our results indicate that HP1021 modulates *H. pylori's* response to oxidative stress. HP1021 controls the transcription of 497 genes, including 407 genes related to response to oxidative stress. 79 proteins are differently expressed in the HP1021 deletion mutant. HP1021 controls typical ROS response pathways (*katA*, *rocF*) and less canonical ones, particularly DNA uptake and central carbohydrate metabolism. HP1021 is a molecular regulator of competence in *H. pylori*, as HP1021-dependent repression of the *comB* DNA uptake genes is relieved under oxidative conditions, increasing natural competence. Furthermore, HP1021 controls glucose consumption by directly regulating the *gluP* transporter and has an important impact on maintaining the energetic balance in the cell.

*Helicobacter pylori* is a gram-negative, microaerobic bacterium[1] that belongs to the phylum Campylobacterota (formerly Epsilonproteobacteria)[2]. *H. pylori* is an obligate human pathogen and the leading cause of peptic ulcers, gastric lymphoma, and gastric adenocarcinoma, and the second leading cause of death from cancer worldwide[3]. Moreover, *H. pylori* is considered by WHO as a species that poses a threat to human health and for which new antibiotics are urgently needed[4].

*H. pylori* inhabits the stomach's harsh environment. The complex response to environmental conditions crucial for *H. pylori* survival and pathogenesis, such as acidic pH, nutrient and metal ion availability or response to immune cells, has been relatively well-established[5,6]. The *H. pylori* genome encodes only 17 transcription regulators, including

three sigma factors, which form a regulatory, often overlapping network that helps *H. pylori* to maintain homeostasis and respond to the environmental signals[7,8]. Moreover, it has been recently proposed that post-transcriptional regulation plays a significant role in controlling *H. pylori* gene expression. In the *H. pylori* 26695 strain carrying 1576 ORFs, 1907 transcription start sites (TSS) were identified, out of which more than 900 were assigned as non-coding RNAs (ncRNAs), including small RNAs (sRNAs) and anti-sense RNAs (asRNAs)[9,10]. Significant overlap in regulating gene expression at least at two levels (transcription and post-transcription) leads to poor understanding of many *H. pylori* processes, particularly response to oxidative stress.

Oxidative stress triggered by reactive oxygen species (ROS) affects pathogens during infection[11]. ROS are produced endogenously

[1]Department of Microbiology, Hirszfeld Institute of Immunology and Experimental Therapy, Polish Academy of Sciences, Wrocław, Poland. [2]Department of Molecular Microbiology, Faculty of Biotechnology, University of Wrocław, Wrocław, Poland. [3]Department of Biochemistry and Molecular Biology, Wrocław University of Environmental and Life Sciences, Wrocław, Poland. [4]Quadram Institute Biosciences, Norwich Research Park, Norwich, UK. [5]Bavarian Center for Biomolecular Mass Spectrometry (BayBioMS), Technical University of Munich (TUM), Freising, Germany. [6]Department of Biological Safety, National Reference Laboratory for Campylobacter, German Federal Institute for Risk Assessment, Berlin, Germany. ✉e-mail: anna.pawlik@hirszfeld.pl

in bacteria as by-products of aerobic metabolism or exogenously, e.g., by other bacteria or the host's immune system[11]. ROS are highly toxic to cells because they induce damage to proteins, lipids, cofactors of enzymes and nucleic acids. Therefore, bacteria produce defence factors that help to neutralise ROS, e.g., superoxide dismutase (Sod), catalase (Kat)[12] or low molecular thiols (e.g., glutathione, mycothiol, bacillithiol)[13]. However, recent data indicate that classical pathways of ROS detoxification are aided by redirecting metabolic fluxes from the production of the pro-oxidative metabolite, NADH, towards increasing the antioxidative metabolite, NADPH, ultimately increasing cellular redox potential[14]. In particular, non-stressed cells inhibit their oxidative pentose phosphate pathway (PPP) by the redox cofactor NADPH inactivating the glucose-6-phosphate dehydrogenase (G6PDH). During oxidative stress, this inhibition is relieved to increase the pathway flux to detoxify ROS[15].

*H. pylori* response to oxidative stress is still enigmatic. On the one hand, the pathogen possesses a limited repertoire of ROS-combating enzymes or compounds (details about *H. pylori* response to oxidative and nitrosative stress are reviewed elsewhere[16]). In particular, *H. pylori* lacks the general stress response regulator RpoS and glutathione–glutaredoxin reduction system (GSH/Grx). The bacterium encodes proteins that directly detoxify ROS or regenerate ROS-modified proteins (e.g., KatA, SodB, AhpC), protect molecules from damage (e.g., NapA) or repair damaged molecules (e.g., MsrAB and MutY). Multiple regulators control genes encoding oxidative stress response proteins. For example, *katA* is controlled by the transcriptional regulators Fur, ArsR and HP1021[17–20], *sodB* is primarily controlled by Fur[21,22] and HsrA[23]. On the other hand, *H. pylori* uses non-classical strategies to combat or even benefit from oxidative stress. *H. pylori* can take up antioxidant ergothioneine (EGT, thiourea derivative of histidine), which, based on studies in mice, may facilitate *H. pylori* infection[24]. It has been recently shown that *H. pylori* senses and migrates towards HClO⁻ rich niches characteristic of inflamed tissues–an efficient source of nutrients, especially iron[25]. Nitrosative stress is highly toxic to *H. pylori*; therefore, the bacterium reduces NOS2 expression and, consequently, NO production in the host[26]. Moreover, ROS-induced mutations in DNA may increase *H. pylori* genetic variation and adaptability to human hosts, especially during the acute infection phase[25–27]. It was also proposed that upon oxidative stress, the pathogen incorporates large amounts of DNA by active uptake of free foreign DNA, possibly using it as a shield to protect its DNA from oxidative damage[28]. Moreover, *H. pylori* uses DNA uptake and incorporation to evolve during human infection[29]. However, the molecular mechanism controlling DNA uptake has not been discovered thus far.

We have recently shown that an atypical HP1021 response regulator (i.e., not controlled by phosphorylation) acts as a redox switch protein, thereby sensing and transmitting the cell's redox state and triggering the appropriate cell response[30]. It is one of the least characterised *H. pylori* regulatory proteins. HP1021 is not essential for *H. pylori* viability, but knock-out of HP1021 reduces *H. pylori* growth rate[20,31]. We have shown that in strain *H. pylori* N6, HP1021 controls the expression of *fecA3* and *gluP* in response to oxidative stress while activating the expression of *katA* irrespective of the redox conditions[20]. Still, the regulon of HP1021 has not been precisely defined. The available transcriptomic data of *H. pylori* 26695 wild-type and ΔHP1021 strains indicated that HP1021 controls the transcription of 79 genes[32]. However, the transcriptome was mapped by microarray technology, which has limited qualitative and quantitative capacities compared to RNA-seq. Moreover, the transcriptome was only defined under microaerobic growth conditions and did not represent transcriptional changes induced by oxidative stress.

In this work, using a comprehensive approach, we deciphered the HP1021 regulon. We studied how HP1021 helps *H. pylori* respond to oxidative stress using transcriptomic, proteomic, and DNA-protein interaction analyses. Finally, our studies proved that HP1021 decides

about *H. pylori* response to oxidative stress and directly controls both typical ROS response pathways and less canonical ones, i.e., DNA uptake and central carbohydrate metabolism.

## Results and discussion

### Identification of HP1021 binding sites on *H. pylori* genome

The exact positions of only a few HP1021 genomic targets have been identified thus far, either by ChIP-PCR or EMSA[20,32,33]. Therefore, to identify genome-wide HP1021 targets and to reveal whether the targets are different under microaerobic and aerobic conditions (i.e., optimal and oxidative stress conditions, respectively), we applied a ChIP-seq approach. *H. pylori* N6 wild-type (WT) and ΔHP1021 liquid cultures at the logarithmic growth phase were incubated under microaerobic (WT, ΔHP1021) or aerobic stress conditions (WTS, ΔHP1021S), cross-linked, sonicated, and HP1021 protein-DNA complexes were immunoprecipitated (IP) with a polyclonal anti-HP1021 antibody[33]. A minimum of 2 million reads, with optimal mapping performances (>90%) for each sample and biological replicate were obtained and used to generate the genome-wide binding profiles (Fig. S1a). Combining edgeR and MACS3 algorithms, we identified 100 putative binding sites (Supplementary Data 1). The binding sites located between positions −250 and +250 relative to a transcription start site (TSS) determined previously[10] in the *H. pylori* 26695 strain were associated with a promoter region. However, it should be noted that TSS sites might differ between *H. pylori* strains (e.g., different TSS sites in *htrA-ispDF-HP1021* operon in the 26695 and G27 strains[6]). Thus, we expected some differences in TSS sites assigned for the 26695 strain compared to the N6 strain used in our analysis. Nonetheless, 84 of 100 identified HP1021 binding sites were located near promoter regions, which included previously identified *katA*, *gluP*, *hyuA*, and *fecA3* binding sites[20,32,33] (Supplementary Data 2). The HP1021 protein bound to the same genomic regions in *H. pylori* WT and WTS cells (Fig. S1a, b). It suggests that HP1021 binds the same genomic regions regardless of conditions (microaerobic or under oxidative stress) and implicates the putative mechanism of HP1021 activity regulation similar to that shown for OxyR[34,35]. In particular, HP1021 becomes modified at cysteine residues upon oxidative stress[20], which possibly triggers structural changes in HP1021, remodels the HP1021-DNA complex and activates or inhibits transcription. However, in the case of four binding sites, the number of reads decreased in WTS compared to WT cells, indicating that in some cases, oxidative stress promoted dissociation of the HP1021-DNA complexes. For example, the binding of HP1021 to the promoter region of *vacA* and dissociation of the p*vacA*-HP1021 complex upon oxidative stress in vivo was detected by ChIP-seq and confirmed by ChIP-qPCR (Fig. S2a and S2d); HP1021 bound p*vacA* probe in vitro (Fig. S2e). The lack of HP1021 in ΔHP1021 cells and the dissociation of HP1021 upon oxidative stress in WTS cells possibly trigger increased *vacA* transcription (Fig. S2a–c).

To conclude, we detected 100 binding sites of HP1021 on the *H. pylori* N6 chromosome, most of which were promoter-located, likely affecting gene transcription. However, since many *H. pylori* genes are organised in operons[10], the number of genes regulated by HP1021 is possibly higher.

### Influence of HP1021 on *H. pylori* transcriptome

To identify the HP1021 regulon and to elucidate its role in *H. pylori* response to oxidative stress, RNA-seq transcriptome analysis of the *H. pylori* N6 wild-type and ΔHP1021 mutant strains was performed under microaerobic growth (WT, ΔHP1021) and upon aerobic stress (WTS, ΔHP1021S). We expected to primarily detect the direct, immediate transcriptional response after a short 25-min oxidative stress because induction of protein synthesis, which could trigger a possible downstream response, was shown to be relatively slow in *H. pylori*[36,37] (see also below MS proteomic results). Moreover, it is also known that oxidative stress usually reduces the translation rate[38,39].

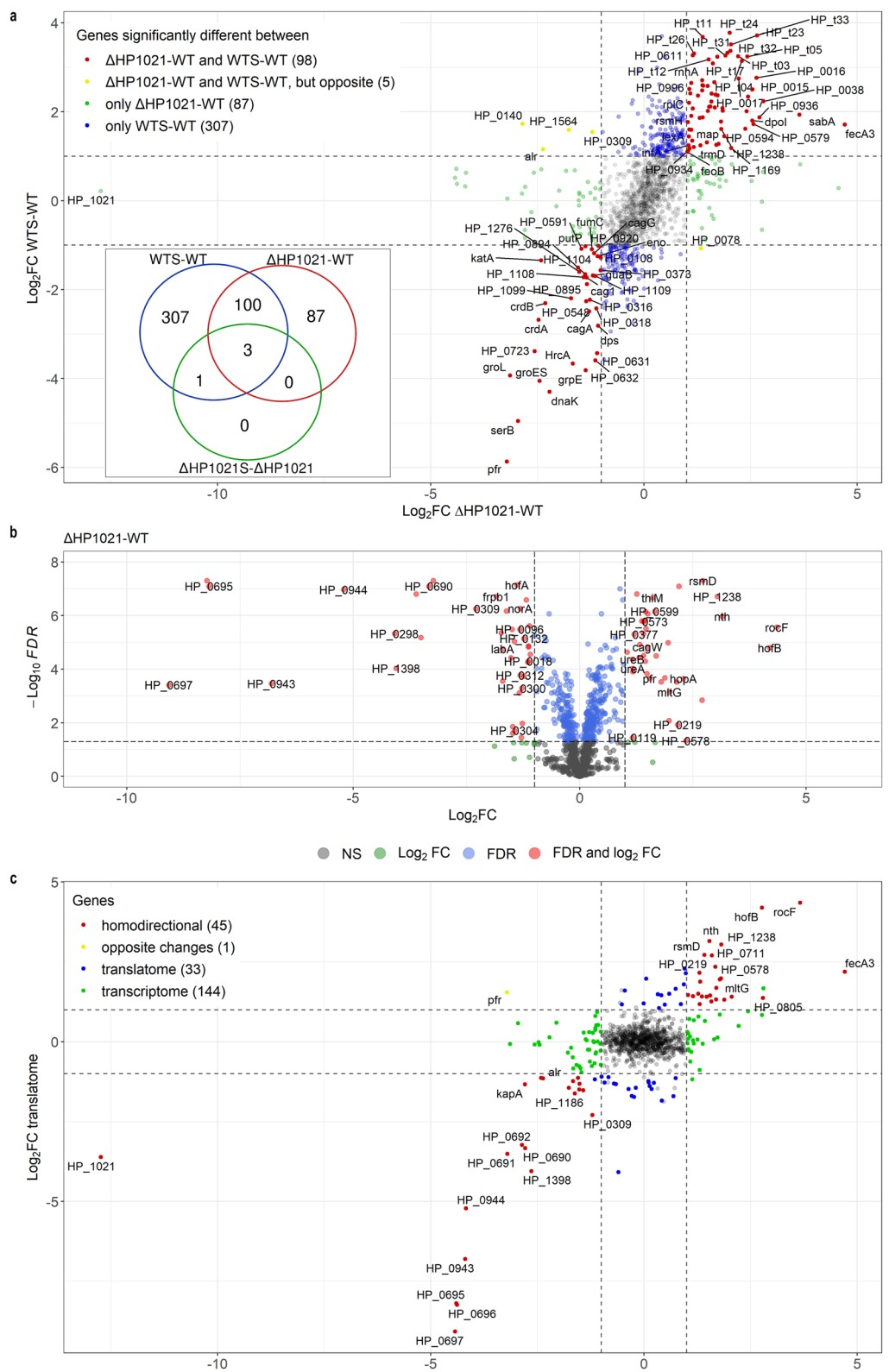

The biological replicates were reproducible, and the grouping of genotypes and/or treatments was preserved (Fig. S3a, b). The transcription of 411 genes out of 1506 *H. pylori* genes was significantly affected (up- or down-regulated) by oxidative stress in WTS compared to WT cells (Fig. 1a, Fig. S4a and Supplementary Data 2). A comparison of genes transcribed in the ΔHP1021 and WT cells revealed 190 differentially transcribed genes, which included 103 genes whose

transcription was affected by oxidative stress in WTS cells (Fig. 1a, Fig. S4b and Supplementary Data 2, the HP1021 gene was not counted as differentially expressed). In contrast to *H. pylori* wild-type cells, in which the transcription of nearly 30% of genes was affected by oxidative stress, the transcription of only four genes significantly changed under oxidative stress in the *H. pylori* ΔHP1021S cells (Fig. S4c and Supplementary Data 2). The transcription of three of these genes

**Fig. 1 | HP1021 controls the expression of *H. pylori* N6 genes under microaerobic and aerobic conditions (5% and 21% O$_2$, respectively). a** The overview of the gene regulation mediated by HP1021 in *H. pylori* N6 revealed by RNA-seq. Genes whose transcription significantly changed (|log$_2$FC| ≥ 1; FDR ≤ 0.05) are depicted by red dots (WTS-WT and ΔHP1021-WT), yellow dots (WTS-WT and ΔHP1021-WT, but opposite), green dots (ΔHP1021-WT) and blue dots (WTS- WT). Gray dots correspond to genes whose transcription was not significantly changed (NS) in WTS-WT or ΔHP1021-WT. The inlet figure shows a Venn diagram presenting the number of differentially transcribed genes in the analyzed strains and conditions. **b** Volcano diagram of proteins differentially expressed in the ΔHP1021 mutant strain compared to the wild-type (WT) strain (ΔHP1021-WT). Green dots correspond to genes with |log$_2$FC| ≥ 1 and FDR ≥ 0.05; blue dots correspond to genes with |log$_2$FC| ≤ 1 and FDR ≤ 0.05; red dots correspond to genes with |log$_2$FC| ≥ 1 and FDR ≤ 0.05; gray dots correspond to genes that were not significantly changed (NS). **c** Correlation between gene expression of ΔHP1021 and WT cells at the transcription and proteome levels. The x-axis corresponds to the RNA-seq data (transcriptome), and the y-axis to the proteomics data (translatome). Red dots represent homodirectional genes–upregulated or down-regulated–at the transcriptome and translatome levels. Yellow dots represent opposite changes for comparing transcriptome and translatome. Blue dots represent genes that changed only at the transcription level, while green dots represent genes that changed only at the translatome level. Gray dots correspond to genes whose transcription was not significantly changed (NS, FDR ≤ 0.05). **a–c** Values outside the black dashed lines indicate a change in the expression of |log$_2$FC| ≥ 1. **a, c** Numbers of differentially expressed genes in the indicated strains and conditions are given in parentheses. The HP1021 gene was not counted as differentially expressed.

---

differed between ΔHP1021 and WT cells under microaerophilic conditions (e.g., *pfr* (*HPO653*), discussed below). Thus, they belong to HP1021 regulon, but additional factors may control their response to oxidative stress. Nonetheless, these results indicated that the *H. pylori* ΔHP1021 strain almost did not respond to oxidative stress. Moreover, the results indicated that 307 genes similarly transcribed in ΔHP1021 and WT strain under microaerobic conditions, which transcription changed in WTS cells but was not affected by oxidative stress in ΔHP1021S cells, also belong to HP1021 regulon (Fig. 1a). Thus, 407 genes differently transcribed in WTS cells whose transcription did not change in the ΔHP1021S strain constitute the HP1021-dependent oxidative stress regulon (Fig. 1a). There were also 87 genes whose transcription did not change under oxidative stress in WTS cells, but their transcription differed between ΔHP1021 and WT strains (Fig. 1a). These genes were possibly activated or inactivated as a second-line response in ΔHP1021 cells to reach homeostasis under a transcriptional state highly different from that of the WT cells. Thus, these genes can be included in the HP1021 regulon independent of oxidative stress.

To conclude, the HP1021 regulon includes 497 genes (Fig. 1a). The transcriptional regulation of 407 of these genes is related to oxidative stress, while the transcription of 87 genes is related to unknown conditions. HP1021 controls the transcription of 3 genes, which response to oxidative stress is possibly controlled by other or additional regulators. Moreover, 48 of the detected binding sites of HP1021 are associated with the genes or operons differentially transcribed in the ΔHP1021 and/or under oxidative stress (Supplementary Data 2). These genes/operons are putatively directly controlled by HP1021.

### Influence of HP1021 on *H. pylori* translatome
To further validate RNA-seq analyses, a proteomic approach was adopted to analyse differential protein occurrence. Similarly to RNA-seq, wild-type N6 and ΔHP1021 mutant strains were compared under microaerobic and aerobic conditions. Due to the additional time necessary for protein synthesis, bacteria were stressed longer than in transcriptome analyses, i.e., 60 and 120 min. The bacterial lysate was collected, and quantitative proteomics using liquid chromatography-based tandem mass spectrometry (LC-MS/MS) was performed to analyse the whole-cell proteomes. The biological replicates were reproducible, and the grouping of genotypes and/or treatments was preserved (Fig. S3c).

Altogether, 1139 proteins were identified. 79 proteins were differentially expressed between WT and ΔHP1021 strains under microaerobic conditions (Fig. 1b and Supplementary Data 2). Out of all 190 genes whose transcription changed in ΔHP1021 compared to the WT strain, the level of 46 proteins changed in ΔHP1021 (Fig. 1c), the level of 74 proteins did not change, while 70 proteins were not detected by MS. Changes in transcription and protein levels correlated for 45 genes (red dots in Fig. 1c and Supplementary Data 2). The level of one protein, Pfr ferritin (HP0653), was opposite to that expected from RNA-seq results (Fig. 1c, yellow dot and Supplementary Data 2).

Upon oxidative stress, the levels of only three proteins changed in wild-type cells (Fig. S5a, b). In the ΔHP1021 mutant, only the endonuclease III Nth (HP0585) protein level was affected by oxidative stress (Fig. S5c, d, discussed below). The discrepancy between the vast transcriptomic response to oxidative stress (transcription of 411 genes changed in WTS cells compared to WT cells) and the mainly unchanged level of all cellular proteins under stress might be explained by a general mechanism of translation inhibition in bacteria upon oxidative stress[40]. Nonetheless, RNA-seq results may show how bacteria prepare for response to stress or recovery when the stressor gets milder or is away, allowing translation[38,39]. Different results in transcriptomic and proteomic studies might additionally be due to the high post-transcriptional modification processes in *H. pylori*[9]. Nonetheless, transcriptional changes and overall final protein levels correlated across multiple genes showing the direction of cell adaptation for the lack of HP1021 protein and possibly a response to oxidative stress controlled by HP1021.

### HP1021 controls many cellular pathways and processes
The significant number of genes assigned to the HP1021 regulon suggests that HP1021 regulates many pathways and processes. Indeed, the performed functional analysis (eggNOG) indicated that the expression of genes belonging to many Clusters of Orthologous Groups (COG), involved in response to oxidative stress as well as those unrelated to cell's response to oxidative stress, was affected by the lack of HP1021 at the transcription and/or translation steps (Fig. S6a–c). In particular, significantly affected functional groups (P ≤ 0.05) were connected with (i) cell energy production and conversion, (ii) translation, (iii) cell wall and envelope structure and biogenesis, (iv) molecular chaperons (v) lipid metabolism and (vi) intracellular trafficking, secretion and transport. In all analyses, many coding sequences of unknown functions were significantly affected.

Due to the high number of gene expression changes, we chose ROS/RNS response, DNA uptake, and glucose metabolism to analyse HP1021-dependent control in more detail.

### HP1021 is involved in *H. pylori* response to oxidative stress
RNA-seq analysis revealed that, in contrast to the wild-type strain, the ΔHP1021 deletion mutant does not respond to oxidative stress at the transcription level, except for four genes which were up- or down-regulated upon oxidative stress (Fig. S4c and Supplementary Data 2). It indicated that HP1021 decides about *H. pylori* response to oxidative stress, regulating genes belonging to classical response to ROS and RNS[16,41] (Supplementary Data 2 and Supplementary Data 3) and non-canonical response pathways, including DNA-uptake and metabolic changes (see below). It should be noted that CrdSR two-component system regulates *H. pylori* response to RNS[42,43]. Based on ChIP-seq, RNA-seq, and MS analyses, we conclude that two enzymes, *katA* (HP0875) (Fig. S7a, b; Supplementary Data 2 and Supplementary Data 3) and *rocF* (HP1399) (Fig. S8; Supplementary Data 2 and Supplementary Data 3), were directly controlled by HP1021 in WT strain

under microaerobic conditions, which confirmed the previous results[20,32]. Consistency of decreased transcription and protein levels of KatA, KapA (KatA accessory protein HP0874, in an operon with *katA*), and increased expression of RocF in the ΔHP1021 strain suggested no additional post-transcriptional control of these genes under the conditions tested. Consequently, N6, 26696 and P12 ΔHP1021 mutant strains were characterised by lower catalase activity in ΔHP1021 strains than in WT strains (Fig. S7c–e and[20]). We did not analyse arginase activity in the ΔHP1021 strain. RocF plays an important role in metabolism in bacteria[44] but is also considered a virulence factor (e.g., in *Xanthomonas oryzae* pv *oryzae* increasing exopolysaccharide synthesis, biofilm formation and $H_2O_2$ resistance)[45]. Arginine is an amino acid essential for *H. pylori* viability. RocF arginase, which decomposes arginine to urea and $CO_2$, was proposed as important for urease-independent acid protection and in RNS stress response[46]. However, the *rocF* deletion mutants also exhibited lower serine dehydratase activity[46]. Serine dehydratase (Sdh, HP0132) deaminates serine to pyruvate and ammonium; pyruvate plays a central role in *H. pylori* carbon metabolism[47,48]. Lower Sdh activity correlated with less-effective mice colonisation by *H. pylori*, which eventually suggested a relationship between *H. pylori* metabolism and colonisation efficiency[49]. In ΔHP1021 RocF protein level was 20–fold increased, which suggested that it plays an important role in compensating for the lack of HP1021. However, further studies are required to reveal the mechanism of RocF-mediated *H. pylori* response to oxidative stress.

The regulation of other ROS/RNS detoxification genes by HP1021 seems more complex and may involve additional post-transcriptional control. For example, ferritin (Pfr, HP0653) transcription was downregulated more than 10-fold in ΔHP1021 and WTS cells and almost 3-fold in ΔHP1021S compared to ΔHP1021. The results were counterintuitive because ferritin is required to sequester free iron to protect the cell from damage caused by the Fenton reaction under oxidative stress[50]. However, the LC-MS/MS analysis indicated that the protein level increased in ΔHP1021 approximately three-fold when compared to the WT strain (fold change (FC) of 2.92), which suggested that in *H. pylori*, the mechanism of iron sequestration in response to oxidative stress is maintained. Moreover, it indicates that *pfr* expression is controlled not only by Fur[51] and HP1021 but also post-transcriptionally, possibly by HPnc3280 RNA, as already suggested[10] (Supplementary Data 2 and Supplementary Data 3). Due to the inhibition of protein synthesis during oxidative stress and taking into account high post-transcriptional control of gene expression in *H. pylori*, it is difficult to determine the HP1021-controlled cell response univocally. Interestingly, the only protein whose level increased in the ΔHP1021 strain upon stress, endonuclease III (Nth), involved in repairing oxidative pyrimidine damage, is also possibly post-transcriptionally controlled (transcription did not change, while protein level increased two-fold in ΔHP1021S cells compared to ΔHP1021), which was also indicated[10]. Nonetheless, the data suggest that the *H. pylori* response to oxidative stress is complex and needs further investigation.

## HP1021 controls the first step of DNA uptake by *H. pylori*

It has recently been shown that *H. pylori* takes up DNA in response to oxidative stress[28]. However, the mechanism of DNA uptake regulation has not been studied at the molecular level in *H. pylori*. DNA uptake is a two-step process[50], with the first step of transport over the outer membrane being mediated by a ComB type IV secretion system (T4SS)[52]. In the first step, double-stranded DNA (dsDNA) is taken up from the environment to the periplasm, while in the second step, it is unwound and transported as single-stranded DNA (ssDNA) into the cytoplasm. The genes encoding ComB T4SS engaged in the first step of DNA uptake are grouped in two gene clusters, *comB2-comB4* and *comB6-comB10* (*HP0015-HP0017* and *HP0037-HP0041*, respectively), while the genes involved in the second step are scattered on the

chromosome. The expression level of the ComB complex was proposed to be a limiting factor for transformation efficiency[53]. ChIP-seq analysis revealed binding sites of HP1021 upstream of *comB2* and *comB8* genes (Fig. 2a, b, Supplementary Data 3). Transcriptome data indicated that the expression of genes of these two operons, except *comB*6, was upregulated in ΔHP1021 and WTS cells (between 3 to 7 fold) compared to WT cells (Fig. 2c, d). Proteomic analyses only identified ComB4 and ComB8, which can be explained by the membrane localisation of most proteins that comprise the ComB transport system. In line with the transcriptomics data, these two proteins were three times more abundant in ΔHP1021 mutant compared to the WT, even though this difference was not significant according to our criteria (ComB4: FC of 3.05, FDR = 0.45; ComB8: FC of 3.18, FDR = 0.053). We performed additional experiments to analyse whether HP1021 directly regulates DNA uptake by *H. pylori*. Using the RT-qPCR, we analyzed the transcription of *comB8* in the N6 strain background comparing WT, ΔHP1021 and COM/HP1021 under microaerobic and oxidative stress conditions (Fig. 2e). In the WT and COM/HP1021 strains, the transcription of *comB8* was upregulated under oxidative stress compared to their expression in the non-stressed cells (FC of $8.4 \pm 2.5$ and $4.7 \pm 1$, respectively). In the ΔHP1021 strain under microaerobic conditions, the transcription of *comB8* was upregulated compared to the non-stressed WT strain (FC of $13.7 \pm 6.4$) and did not significantly change upon oxidative stress (FC of $19.5 \pm 8.5$). To confirm the interaction of the HP1021 protein with the *comB8* promoter region in the N6 wild-type strain, we applied ChIP-qPCR and EMSA. In ChIP-qPCR, the binding of HP1021 to the *comB8* promoter region was $110.4 \pm 11$-fold enriched relative to the non-antibody control indicating a high affinity of HP1021 to the *comB8* promoter region. Under oxidative stress, the binding affinity to the promoter region was similar to the non-stressed conditions (FC of $110.78 \pm 6.4$ relative to the non-antibody control) (Fig. 2f). The *HP1230* gene was used as a negative control to represent the lack of interactions with HP1021 (FC of $10.6 \pm 1.4$ and $10.22 \pm 1.3$ fold enrichment in stress relative to the non-antibody control). The interaction between HP1021 and the promoter region of *comB* (p*comB8*) was confirmed by EMSA (Fig. 2g). This region contains two sequences similar to the consensus sequence of HP1021 box[54] (GGTTGCA and GGTTTCT), and the affinity of HP1021 to p*comB8* and *oriC2*, which contains three HP1021 boxes, was comparable.

To further analyse whether HP1021 controls DNA uptake by *H. pylori* cells, we determined the number of cells with active DNA uptake complexes in WT and ΔHP1021 strains under microaerobic and aerobic conditions, as described previously[55]. Active DNA uptake was monitored by tracking Cy-3 labeled λ DNA import in single cells using fluorescence microscopy (Fig. 2h). Under microaerobic conditions, uptake of DNA into a DNase resistant state was detected in $23.54\% \pm 7$ of the N6 WT cells (Fig. 2i). The number of cells importing DNA/being competent increased under aerobic conditions up to $40.72\% \pm 6.2$, confirming previous results[55]. In the N6 ΔHP1021 strain, active DNA uptake was detected in almost every cell regardless of the conditions (in $94.95\% \pm 1.9$ of non-stressed cells and $96.24\% \pm 2.9$ of stressed cells). In COM/HP1021 strain, the number of competent cells under microaerobic conditions was higher than in the WT cells ($36.98\% \pm 3.8$). However, similarly to WT cells, the number of COM/HP1021 cells importing DNA increased approx. 2-fold under aerobic conditions ($54.21\% \pm 8.8$). Since *H. pylori* strains are highly variable[56], we used a set of P12 strains (WT, ΔHP1021, COM/HP1021; Table S1) to analyse whether the mechanism of HP1021-dependent DNA-uptake control is universal. The results were similar to that obtained in the N6 strain series (Fig. S9a, b). As mentioned above, DNA uptake is a two-step process[52]. Only the operons encoding proteins involved in the DNA transport to the periplasm but not directly to the cytoplasm were activated. Thus, we wanted to analyse the transformation rate in *H. pylori* N6 cells under microaerobic and oxidative stress conditions. We used the *rpsL* (A128G) point mutation marker to measure the transformation rate[57].

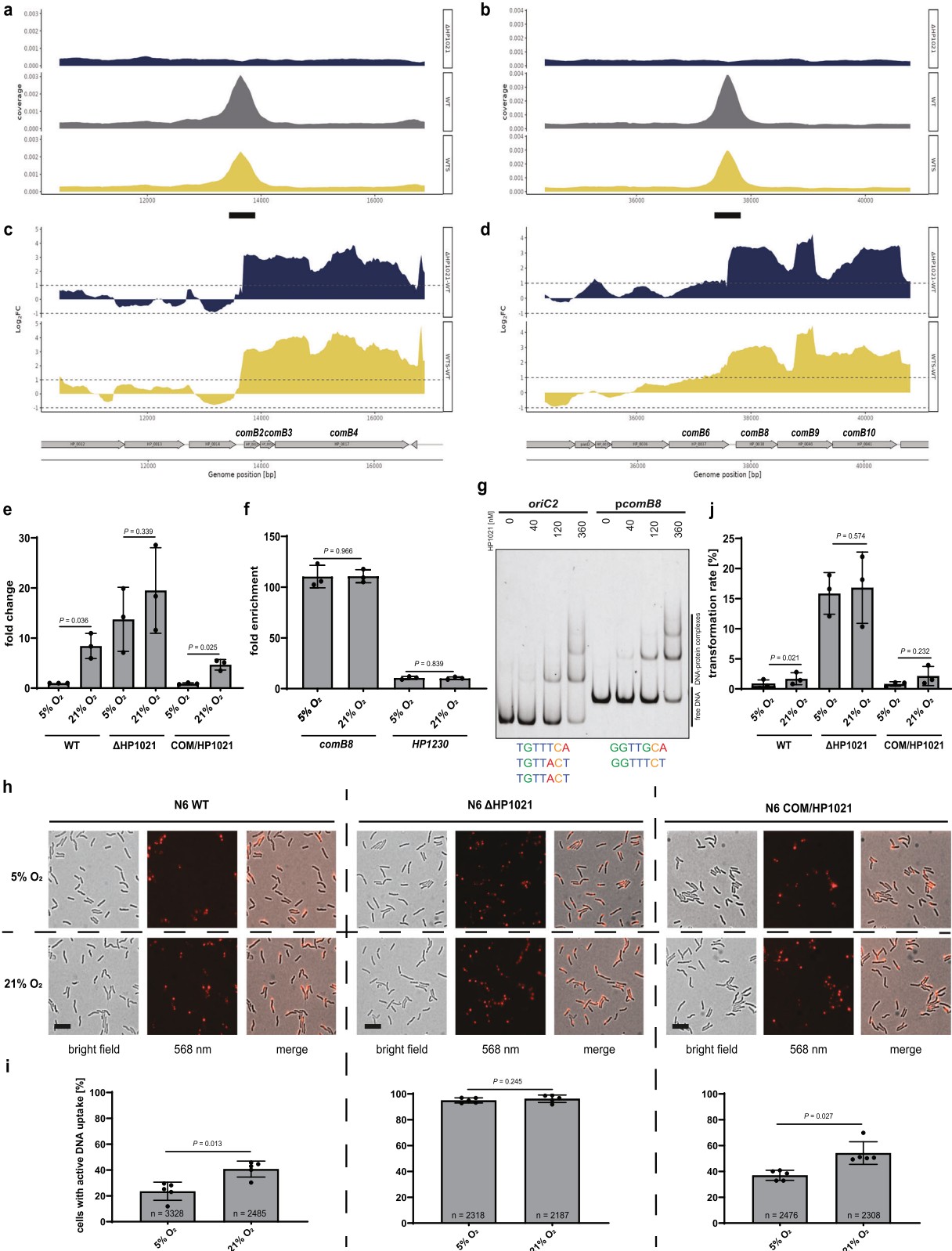

After DNA uptake and recombination with a chromosome, *rpsL* (A128) point mutation confers streptomycin resistance allowing us to measure the natural transformation rate. Under stress conditions, the transformation rate of WT and COM/HP1021 strains was approx. three times higher than under microaerobic conditions (1.7% ± 1 in WT and 2.1% ± 1.6 in COM/HP1021) (Fig. 2j). The established transformation rate under microaerobic conditions in the strain ΔHP1021 (15.86% ± 3.4)

was approx. 20 times higher than in WT (0.7% ± 0.7) and COM/HP1021 (0.8% ± 0.4) strains, and the transformation rate did not change upon oxidative stress (16.76% ± 5.9). Consistently, the number of transformants obtained under microaerobic or aerobic conditions in ΔHP1021 was approx. eight times higher compared to the stressed WT strain. Since DNA uptake to the periplasm occurred in nearly all cells of the ΔHP1021 strain, the transformation rate was limited at a downstream

**Fig. 2 | HP1021 controls the expression of *comB2-comB4* and *comB6-comB10* gene clusters and DNA uptake. a, b** ChIP-seq data profiles. Read counts were determined for *H. pylori* N6 WT, WTS, and ΔHP1021 strains. y-axis, the coverage of the DNA reads; x-axis, the position of the genome; a thick black line under the x-axis, the main peak of the binding site. **c, d** RNA-seq data profiles. The genomic locus for *H. pylori* N6 WT, WTS, and ΔHP1021 strains with the WTS-WT and ΔHP1021-WT expression comparison; values above the black dashed lines indicate a change in the expression of |log₂FC| ≥ 1; FDR ≤ 0.05. **e** RT-qPCR analysis of the transcription of *comB8* in *H. pylori* N6 cultured under microaerobic and aerobic conditions. The results are presented as the fold change compared to the WT strain. **f** ChIP fold enrichment of DNA fragment in *comB8* by ChIP-qPCR in *H. pylori* N6 cultured under microaerobic and aerobic conditions. The *HP1230* gene was used as a negative control not bound by HP1021. **g** EMSA analysis of HP1021 binding to the p*comB8* region in vitro. EMSA was performed using the FAM-labeled DNA fragments and recombinant Strep-tagged HP1021. The *oriC2* DNA fragment was used as a control. The HP1021 boxes are shown below the gel image. **h** Analysis of DNA uptake by *H. pylori* N6. Bright field, fluorescent, and merged images of *H. pylori* WT and mutant strains after 15 min of Cy3 λ DNA uptake under microaerobic and aerobic conditions. The scale bar represents 2 μm. **i** Quantitative analysis of λ-Cy3 DNA foci formation in *H. pylori* under microaerobic and aerobic conditions. **j** Analysis of HP1021 influence on transformation rate in *H. pylori* N6 using the *rpsL* cassette in *H. pylori* N6 WT and mutant strains under microaerobic and aerobic conditions. **e, f, i–j** Data have depicted as the mean values ± SD. Two-tailed Student's t-test determined the P value. **e, f, j** n = 3 biologically independent experiments. **i** n = number of cells examined over 5 independent biological experiments. **g, h** Digital processing was applied equally across the entire image. Source data are provided as a Source Data file.

process. Likewise, most of the genes involved in DNA transfer to the cytoplasm and/or homologous recombination (e.g., *recJ, priA, dnaN, dnaX, recR, ruvB, polA, comEC, comF, comH*) were unchanged (Supplementary Data 2 and Supplementary Data 3).

To conclude, our results indicated that HP1021 controls DNA uptake and represses the *comB2-comB4* and *comB8-comB10* genes under microaerobic conditions, limiting natural transformation to a certain level. In this way, our work identified the molecular regulator of competence development in *H. pylori*. The natural competence of *H. pylori* is considered the basis for a large diversity of *H. pylori* strains[29], promoting *H. pylori* chronic infection[58]. The question remains as to why oxidative stress in *H. pylori* stimulates the DNA uptake process to an exceptionally high level. A tempting hypothesis is a dual-use mechanism: imported DNA, acting as a quencher to accept ROS attack, is later used to increase *H. pylori* diversity and might also be degraded and reused as an energy source or as building blocks (e.g., ribonucleotides). As we present below, the demand for ribonucleotides increases under oxidative stress.

## HP1021 controls glucose uptake and energy homeostasis

*H. pylori* metabolises different carbon sources, including glucose[59,60]. However, whether and how *H. pylori* benefits from glucose utilisation has not been fully explained. Recent comprehensive studies on central carbon metabolism confirmed that glucose is mainly metabolised via the Entner–Doudoroff (ED) pathway and non-oxidative pentose phosphate pathway (PPP) while only marginally fueling the tricarboxylic acid cycle (TCA)[48,61]. Consequently, glucose is a main source of building up bacterial cell walls, lipids, and nucleic acids, while other substrates, mainly glutamate and aspartate, feed the TCA cycle and generate ATP[48]. Interestingly, our transcriptome analyses indicated that the level of two out of six rRNA and most tRNA genes were highly (up to 8-fold) upregulated in the wild-type strain under oxidative stress and in ΔHP1021 (Fig. S10c, d and Supplementary Data 3). Moreover, HP1021 binds to promoter regions of tRNA genes (Fig. S10a, b and Supplementary Data 3), which suggests that HP1021 directly controls tRNA synthesis. Since rRNA and tRNA constitute a significant fraction of cellular components, one can assume that upregulation of these genes requires a substantial input of ribonucleotides as building blocks and energy, for which glucose uptake may be necessary. Previous microarray and RT-qPCR studies indicated that HP1021 controls the *gluP (HP1174)* expression[32] increasing *gluP* transcription upon oxidative stress[20]. ChIP-seq analysis revealed that HP1021 bound *gluP* promoter region in vivo (Fig. 3a). In ChIP-qPCR, the binding of HP1021 to the *gluP* promoter region was 331.6 ± 28.1 fold enriched relative to the non-antibody control confirming a high affinity of HP1021 to the *gluP* promoter region (Fig. 3c). Under oxidative stress, the binding affinity to the promoter region was reduced relative to the non-stressed conditions (FC of 234.7 ± 27.7 relative to the non-antibody control), suggesting that HP1021 oxidation leads to destabilisation of HP1021-*gluP* complexes, as shown before in vitro[20]. No binding of HP1021 was

detected to the *HP1230* gene used as a negative control for lack of interactions with HP1021 (as in Fig. 2f). Thus, we conclude that HP1021 directly controls the transcription of *gluP*. RNA-seq analysis confirmed that the transcription of glucose uptake transporter *gluP* was upregulated in ΔHP1021 (FC of 3.4) (Fig. 3b and Supplementary Data 2). The upregulation of transcription in WTS cells was lower than two-fold (FC of 1.8); nonetheless, it was statistically significant (FDR = 0.000014).

To further analyse whether HP1021 controls glucose uptake in *H. pylori*, we determined the glucose consumption by *H. pylori* wild-type, ΔHP1021, and COM/HP1021 strains in liquid culture. We measured how much glucose was diminished from the TSBΔD-FCS medium supplemented with 100 mg/dl glucose during *H. pylori* growth. First, we confirmed that GluP is the only glucose transporter because the *H. pylori* N6 Δ*gluP* strain (Fig. S11c and Table S1) did not consume glucose (Fig. 3d and Fig. S11a). At the same time, the *gluP* complementation mutant COM/*gluP* partially restored glucose consumption. The calculated glucose consumption per *H. pylori* cell measured in mid-logarithmic culture (OD₆₀₀ = 0.7–1.0) indicated that ΔHP1021 consumed approx. 184% glucose compared to the corresponding wild-type strain (Fig. 3d). The glucose consumption in COM/HP1021 strains was similar to that of the wild-type strain. A similar glucose uptake pattern was observed in P12 WT and isogenic ΔHP1021 and COM/HP1021 strains (Fig. 3d and Fig. S11b). Thus, we concluded that HP1021 controls glucose uptake by regulating the transcription of the GluP transporter.

Despite increased glucose consumption, *H. pylori* ΔHP1021 strains grew slower than WT strains (Fig. S11a, b), suggesting they may lack sufficient energy. Indeed, the transcription of many genes encoding tricarboxylic acid cycle (TCA) enzymes was downregulated in the wild-type strain under oxidative stress and/or in ΔHP1021, including succinyl-CoA:3-ketoacid CoA transferase, which protein level was reduced 10-fold (Fig. S12 and Supplementary Data 3). Moreover, the activities of pyruvate:flavodoxin oxidoreductase (PFOR) and 2-oxoglutarate:acceptor oxidoreductase (OOR) enzyme complexes are reduced in the presence of ROS[62], which leads to inhibition of substrate supply into TCA. Furthermore, the genes encoding enzymes providing building blocks for nucleic acids were upregulated in ΔHP1021 strain (PPP *rpiB (HP0574)*: RNA-seq FC of 2.5, MS FC of 3.7; nucleotide recycling *deoB (HP1179)*: MS FC of 2.7) (Fig. S12 and Supplementary Data 3). We measured ATP levels in WT and mutant strains in the mid-logarithmic growth phase to analyse whether energy production and/or consumption was lower in the ΔHP1021 strain. *H. pylori* ΔHP1021 strain had approx. 75% ATP level of the WT strain, while the complementation mutant restored ATP to approx. 107% of the WT strain (Fig. 3e). We cannot distinguish whether the ATP level dropped due to decreased ATP synthesis via the TCA cycle or increased ATP consumption for anabolic reactions. However, we postulate that HP1021 controls glucose uptake and, in the presence of glucose, regulates *H. pylori* metabolic fluxes to maintain the balance between anabolic and catabolic reactions, possibly for efficient stress response.

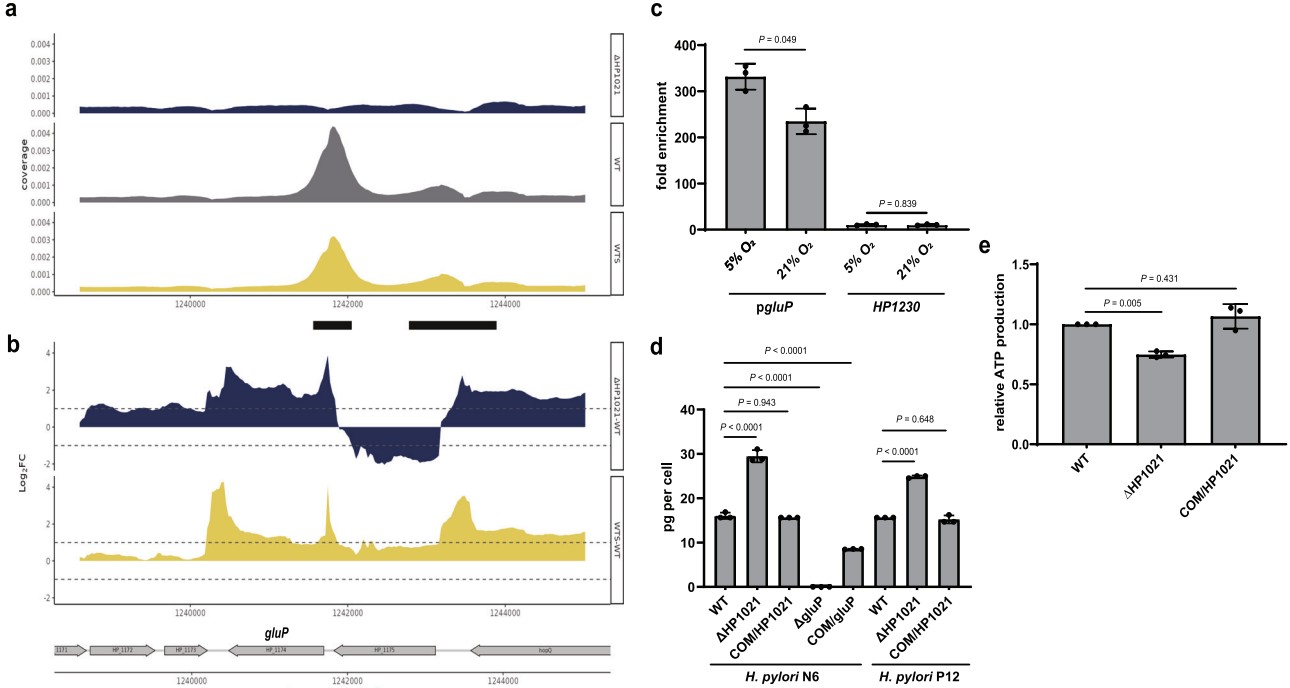

**Fig. 3 | HP1021 controls *gluP* expression and glucose uptake. a** ChIP-seq data profile of the *gluP* gene. Read counts were determined for *H. pylori* N6 WT, WTS and ΔHP1021 strains. The y-axis represents the coverage of the DNA reads, while the x-axis represents the position of the genome (in bps). The main peak of the binding site is marked with a thick black line under the x-axis. **b** RNA-seq data profile of *gluP* gene. The genomic locus for *H. pylori* N6 WT, WTS, and ΔHP1021 strains with the WTS-WT and ΔHP1021-WT expression comparison; values above the black dashed lines indicate a change in the expression of $|\log_2 FC| \geq 1$; FDR ≤ 0.05. **c** ChIP fold enrichment of DNA fragment in *gluP* by ChIP-qPCR in *H. pylori* N6 cells cultured under microaerobic and aerobic conditions (5% and 21% $O_2$, respectively). The

HP1230 gene was used as a negative control not bound by HP1021. Two-tailed Student's t-test determined the P value. **d** Glucose consumption of *H. pylori* N6 and P12 wild-type and mutant strains. Glucose consumption is presented as picograms of glucose used by *H. pylori* cells grown in liquid culture between the inoculation and late logarithmic growth phase. **e** ATP production by *H. pylori* N6 wild-type and mutant cells in the late logarithmic growth phase. **c–e** Data are depicted as the mean values ± SD. n = 3 biologically independent experiments. **d, e** Ordinary one-way ANOVA with Tukey's multiple comparison test determined the P value. Source data are provided as a Source Data file.

To conclude, *H. pylori* response to oxidative stress was controlled by HP1021, which activated the classical and non-canonical response mechanisms. Our research particularly emphasised the role of nucleic acids in *H. pylori* protection against oxidative stress. Similarly to other bacteria, including *Escherichia coli*, *H. pylori* ceased protein synthesis upon oxidative stress. However, unlike *E. coli*[38,39], *H. pylori* upregulates the synthesis of rRNA and tRNA. Moreover, *H. pylori* activated DNA import upon oxidative stress, and nearly all ΔHP1021 strain cells were competent and imported DNA under normal growth conditions. Both processes are counterintuitive when the high sensitivity of nucleic acids to ROS damage is concerned[40,63]. Therefore, we propose that increased RNA synthesis may align with DNA uptake—excessive levels of ribonucleic acids may act as ROS quenchers and protect the cell from damage. RNA synthesis, however, requires building blocks and energy, both of which can be supported by the increased glucose uptake and metabolism redirected by HP1021 towards the pentose phosphate pathway, as published before for other bacteria[15,16].

## Methods
### Materials and culture conditions
The strains, plasmids, and proteins used in this work are listed in Supplementary Table S1. The primers used in this study are listed in Supplementary Table S2. *H. pylori* was cultivated at 37 °C at 140 rpm orbital shaking under microaerobic conditions (5% $O_2$, 10% $CO_2$, and 85% $N_2$) generated by the jar evacuation-replacement method (Anaerobic Gas System PetriSphere). *H. pylori* plate cultures were grown on Columbia blood agar base medium supplemented with 10% defibrinated horse blood (CBA-B). The liquid cultures were prepared in Brucella Broth (BB) (Becton Dickinson), Tryptic Soy Broth (TSB)

(Becton Dickinson) or Tryptic Soy Broth without Dextrose (TSBΔD) (Becton Dickinson), each of which contained 10% fetal bovine serum (FBS) (Biowest) (BB-FBS, TSB-FBS, TSBΔD-FBS, respectively). All *H. pylori* cultures were supplemented with an antibiotic mix[64]. If necessary for selecting *H. pylori* mutants, appropriate antibiotics were used with final concentrations (i) kanamycin 15 μg/ml, (ii) chloramphenicol 8 μg/ml, and (iii) streptomycin 20 μg/ml. *Escherichia coli* DH5α and BL21 were used for cloning and recombinant protein synthesis. The *E. coli* MC1061 strain was used for the plasmid propagation to transform *H. pylori*. If necessary for selecting *E. coli*, appropriate antibiotics were used with final concentrations (i) kanamycin 50 μg/ml and (ii) ampicillin 100 μg/ml.

### RNA isolation
Bacterial cultures (12 ml BB-FBS) of *H. pylori* N6 wild-type and ΔHP1021 strains were grown under microaerobic conditions to $OD_{600}$ of 0.8–1.0. Immediately after opening the jar, 1 ml of the non-stressed culture was added to 1 ml of the RNAprotect Bacteria Reagent (Qiagen), vortexed, and incubated for 5 min at room temperature. In parallel, the cultures were moved to aerobic conditions for 25 min (air atmosphere, 37 °C, 140 rpm orbital shaking). After oxidative stress, samples were collected similarly to non-stressed cells. After 5 min incubation with RNAprotect Bacteria Reagent, bacteria were collected by centrifugation (7000 × g, 5 min, RT). RNA was isolated by GeneJET RNA Purification Kit (Thermo Fisher Scientific; K0731) according to the manufacturer's protocol and treated with RNase-free DNase I (Thermo Fisher Scientific). Next, the subsequent purification by the GeneJET RNA Purification Kit was performed to remove DNase I. A NanoDrop Lite spectrophotometer, agarose gel electrophoresis, and Agilent

4200 TapeStation System were used to determine the RNA quality and quantity. RNA was isolated immediately after bacteria collection, stored at −80 °C for up to one month, and used for RNA sequencing. RNA was isolated from three independent cultures.

## RNA sequencing (RNA-seq)

The prokaryotic directional mRNA library preparation and sequencing were performed at the Novogene Bioinformatics Technology Co. Ltd. (Cambridge, UK). Briefly, the ribosomal RNA was removed from the total RNA, followed by ethanol precipitation. After fragmentation, the first strand of cDNA was synthesized using random hexamer primers. During the second strand cDNA synthesis, dUTPs were replaced with dTTPs in the reaction buffer. The directional library was ready after end repair, A-tailing, adapter ligation, size selection, USER enzyme digestion, amplification, and purification. The library was checked with Qubit, RT-qPCR for quantification, and a bioanalyser for size distribution detection. The libraries were sequenced with the NovaSeq 6000 (Illumina), and 150 bp reads were produced.

## RNA-seq analysis

The 150 bp paired reads were mapped to the *H. pylori* 26695 genome (NC_000915.1) using Bowtie2 software with *local* setting (version 2.3.5.1)[65,66] and processed using samtools (version 1.10)[67], achieving more than $10^6$ mapped reads on average. Differential analysis was performed using R packages Rsubread (version 2.10), and edgeR (version 3.38)[68,69], following a protocol described in[70]. Genes rarely transcribed were removed from the analysis (less than 10 mapped reads per library). Obtained count data was normalised using the edgeR package, and a quasi-likelihood negative binomial was fitted. Differential expression was tested using the glmTtreat function with a 1.45-fold change threshold. Only genes with a false discovery rate (FDR) less than 0.05 and $|\log_2 FC| \geq 1$ were considered differentially expressed. Data visualisation with volcano plots and heatmaps wear done using the EnhancedVolcano and tidyHeatmap R packages (version 1.14 and 1.8.1)[71].

## ChIP using a polyclonal HP1021 antibody

Bacterial cultures (70 ml BB-FBS) of *H. pylori* N6 wild-type and ΔHP1021 strains were grown under microaerobic conditions to OD$_{600}$ of 0.8–1.0 and split into 2 sub-cultures of 35 ml each. The first sub-culture was crosslinked with 1% formaldehyde for 5 min immediately after opening the jar. The second culture was crosslinked after 25-min incubation under aerobic conditions (air atmosphere, 37 °C, 140 rpm orbital shaking). The crosslinking reactions were stopped by treatment with 125 mM glycine for 10 min at room temperature. The cultures were centrifuged at 4700 × g for 10 min at 4 °C and washed twice with 30 ml of ice-cold 1 × PBS, followed by the same centrifugation step. Samples were resuspended in 1.1 ml IP buffer (150 mM NaCl, 50 mM Tris-HCl pH 7.5, 5 mM EDTA, 0.5% vol/vol NP-40, 1.0% vol/vol Triton X-100) and sonicated (Ultraschallprozessor UP200s (0.6/50% power, 30 s ON − 30 s OFF, ice bucket)) to reach 100-500 bp DNA fragment size. Next, the samples were centrifuged at 12,000 × g for 10 min at 4 °C. 100 μl of the supernatant was used for input preparation. 900 μl of the supernatant was incubated with 30 μl of Sepharose Protein A (Rockland, PA50-00-0002) (pre-equilibrated in IP buffer) for 1 h at 4 °C on a rotation wheel. The samples were centrifuged 1000 × g for 2 min at 4 °C; the supernatants were incubated with 40 μg of the HP1021 antibody[33] and incubated at 4 °C for 16 h on a rotation wheel. Subsequently, 100 μl of the Sepharose Protein A pre-equilibrated in IP buffer was added to the samples, and the binding reaction was performed for 4 h at 4 °C on a rotation wheel. The samples were centrifuged 1000 × g for 2 min at 4 °C, and the supernatant was discarded. The beads were washed four times with IP-wash buffer (50 mM Tris-HCl pH 7.5, 150 mM NaCl, 0.5% NP-40, 0.1% SDS), twice with TE buffer (10 mM Tris-HCl, pH 8.0; 0.1 mM EDTA), resuspended in 180 μl of TE buffer, and treated

with 20 μg/ml RNase A at 37 °C for 30 min. Next, crosslinks were reversed by adding SDS at a final concentration of 0.5% and proteinase K at a final concentration of 20 μg/ml, followed by incubation for 16 h at 37 °C. The beads were removed by centrifugation 1000 × g for 2 min at 4 °C, and the DNA from the supernatants were isolated with ChIP DNA Clean & Concentrator (Zymo Research). The quality of DNA was validated by electrophoresis in 2% agarose gel, and the concentration was determined with QuantiFluor dsDNA System (Promega). The ChIP-DNA was isolated from three independent *H. pylori* cultures. The HP1021 antibody used in ChIP-seq was raised in rabbits under the approval of the First Local Committee for Experiments with the Use of Laboratory Animals, Wroclaw, Poland (consent number 51/2012).

## ChIP-sequencing

The DNA library preparation and sequencing were performed at the Novogene Bioinformatics Technology Co. Ltd. (Cambridge, UK). Briefly, the DNA fragments were repaired, A-tailed and further ligated with an Illumina adapter. The final DNA library was obtained by size selection and PCR amplification. The library was checked with Qubit and RT-qPCR for quantification and a bioanalyser for size distribution detection. The libraries were sequenced with the NovaSeq 6000 (Illumina), and 150 bp reads were produced.

## ChIP-seq bioinformatic analysis

The 150 bp paired reads were mapped to the *H. pylori* 26695 genome (NC_000915.1) using Bowtie2 software with local setting (version 2.3.5.1)[65,66] and processed using samtools (version 1.10)[67], achieving more than $10^7$ mapped reads on average. Regions differentially bound by HP1021 were identified using R packages csaw (version 1.30), and edgeR (version 3.38)[69,72], following the protocol described in[73]. Briefly, mapped reads were counted using a sliding window (length 100 bp, slide 33 bp) and filtered using local methods from the edgeR package. Only reads with $\log_2$ fold change greater than 2 compared to 2000 bp neighboring region were left for further analysis. Then each window was tested using the QL F-test, and neighboring regions (less than 100 bp apart) were merged. The combined p-value of all merged regions was calculated, and only regions with a false discovery rate (FDR) less than 0.05 were considered differentially bound. All identified regions were further confirmed using the MACS3 (version 3.0.0a6) programme with nomodel settings[74]. Peaks were detected using ΔHP1021 strain files as a control, and only regions with a score higher than 4250 (-$\log_{10}$ from q-value) were considered in subsequent analysis.

## Proteomic sample preparation

Bacterial cultures of wild-type and ΔHP1021 strains were grown in BB-FBS under microaerobic conditions to OD$_{600}$ of 0.8–1.0 and split into three sub-cultures of 10 ml each. The first sub-culture was harvested, washed, and lysed immediately after opening the jar; the second and third sub-cultures were harvested, washed, and lysed after incubation under aerobic conditions (air atmosphere at 37 °C with orbital shaking, 140 rpm) for 60 and 120 min, respectively. The bacterial proteomes were prepared as described previously by Abele et al.[75]. Briefly, cells were harvested by centrifugation at 10,000 × g for 2 min, media were removed, and cells were washed once with 20 ml of 1 × PBS. The cell pellets were suspended and lysed in 100 μl of 100% trifluoroacetic acid (TFA; Roth) for 5 min at 55 °C. Next, 900 μl of Neutralization Buffer (2 M Tris) was added and vortexed. Protein concentration was measured by Bradford assay (Bio-Rad). 50 μg of protein per sample was reduced (10 mM TCEP) and carbamidomethylated (55 mM CAA) for 5 minutes at 95 °C. The proteins were digested by adding trypsin (proteomics grade, Roche) at a 1/50 enzyme/protein ratio (w/w) and incubation at 37 °C overnight. Digests were acidified by the addition of 3% (v/v) formic acid (FA) and desalted using self-packed StageTips (five disks per micro-column, ⌀ 1.5 mm, C18 material, 3 M Empore). The

peptide eluates were dried to completeness and stored at −80 °C. Before the LC-MS/MS measurement, all samples were freshly resuspended in 12 µl 0.1% FA in HPLC grade water, and 25 µg of total peptide amount was injected into the mass spectrometer per measurement. Each experiment was performed using four biological replicates.

## Proteomic data acquisition and data analysis

Peptides were analyzed on a Vanquish Neo liquid chromatography system (micro-flow configuration) coupled to an Orbitrap Exploris 480 mass spectrometer (Thermo Fisher Scientific). 25 µg of peptides were applied onto an Acclaim PepMap 100 C18 column (2 µm particle size, 1 mm ID × 150 mm, 100 Å pore size; Thermo Fisher Scientific) and separated using a two-stepped gradient. In the first step, a 50-minute linear gradient ranging from 3% to 24% solvent B (0.1% FA, 3% DMSO in ACN) in solvent A (0.1% FA, 3% DMSO in HPLC grade water) at a flow rate of 50 µl/min was applied. In the second step, solvent B was further increased from 24% to 31% over a 10-minute linear gradient. The mass spectrometer was operated in data-dependent acquisition (DDA) and positive ionisation mode. MS1 full scans (360 – 1300 m/z) were acquired with a resolution of 60,000, a normalised AGC target value of 100% and a maximum injection time of 50 msec. Peptide precursor selection for fragmentation was carried out using a fixed cycle time of 1.2 s. Only precursors with charge states from 2 to 6 were selected, and dynamic exclusion of 30 s was enabled. Peptide fragmentation was performed using higher energy collision-induced dissociation (HCD) and normalised collision energy of 28%. The precursor isolation window width of the quadrupole was set to 1.1 m/z. MS2 spectra were acquired with a resolution of 15,000, a fixed first mass of 100 m/z, a normalised automatic gain control (AGC) target value of 100% and a maximum injection time of 40 msec.

Peptide identification and quantification were performed using MaxQuant (version 1.6.3.4) with its built-in search engine Andromeda[76,77]. MS2 spectra were searched against the *H. pylori* proteome database derived from *H. pylori* 26695 (NC_000915.1) (containing 1539 protein sequences supplemented with common contaminants (built-in option in MaxQuant)). Trypsin/P was specified as the proteolytic enzyme. The precursor tolerance was set to 4.5 ppm, and fragment ion tolerance to 20 ppm. Results were adjusted to a 1% false discovery rate (FDR) on peptide spectrum match (PSM) level and protein level employing a target-decoy approach using reversed protein sequences. The minimal peptide length was defined as seven amino acids, carbamidomethylated cysteine was set as fixed modification and oxidation of methionine and N-terminal protein acetylation as variable modifications. The match-between-run function was disabled. Protein abundances were calculated using the LFQ algorithm from MaxQuant[78]. Protein intensity values were logarithm transformed (base 2), and a Student t-test was used to identify proteins differentially expressed between conditions. The resulting p-values were adjusted by the Benjamini-Hochberg algorithm[79] to control the false discovery rate (FDR). Since low abundant proteins are more likely to result in missing values, we filled in missing values with a constant of half the lowest detected LFQ intensity per protein. However, if the imputed value was higher than the 20% quantile of all LFQ intensities in that sample, we used the 20% quantile as the imputed value. Only proteins with $|\log_2 FC| \geq 1$ were considered differentially expressed.

## COG categorisation

Differentially expressed genes (DEGs) were functionally annotated using eggNOG-mapper (version 2.13)[80] and assigned functions/categories according to the Cluster of Orthologous Genes (COGs) database[81]. Statistical analysis was carried out using the R statistical software. The frequency of statistically significant genes in RNA-seq/proteomics analysis within COG categories was compared between strains and conditions with the Chi-squared test of independence implemented in R.

## Quantitative polymerase chain reaction

RT-qPCR quantified the mRNA levels of the selected genes. The reverse transcription was conducted using 500 ng of RNA in a 20 µl volume reaction mixture of iScript cDNA Synthesis Kit (Bio-Rad). 2.5 µl of 1:10 diluted cDNA was added to 7.5 µl of Sensi-FAST SYBR No-ROX (Bioline) and 400 nM of forward and reverse primers in a 15 µl final volume. The RT-qPCR program performed as previously described[20]. The following primer pairs were used: P17-P18, *comB8*; P19-P20, *vacA* (Table S2). The relative quantity of mRNA for each gene was determined by referring to the mRNA levels of *H. pylori* 16S rRNA (P15-P16 primer pair). The RT-qPCR was performed for three independent *H. pylori* cultures. The RT-qPCR data were analyzed with CFX Maestro (BioRad) software.

The protein-DNA interactions in the cell in vivo of the selected DNA regions were quantified by ChIP-qPCR. 2.5 µl of 1:10 diluted immunoprecipitation output was added to 7.5 µl of Sensi-FAST SYBR No-ROX (Bioline) and 400 nM of forward and reverse primers in a 15 µl final volume. The ChIP-qPCR was performed using the following programme: 95 °C for 3 min, followed by 40 three-step amplification cycles consisting of 10 s at 95 °C, 10 s at 60 °C and 20 s at 72 °C. The following primer pairs were used: P21-P22, *comB8*; P23-P24, *gluP*; P25-P26, *vacA* (Table S2). The *HP1230* gene was used as a negative control (P27-P28) (Table S2). No-antibody control was used for ChIP-qPCR normalisation, and the fold enrichment was calculated. The ChIP-qPCR was performed for three independent *H. pylori* cultures.

## Determination of the number of cells with active DNA uptake

The DNA uptake assay was performed as previously[55], with minor modifications. λ DNA (*dam-* and *dcm-*; Thermo Fisher Scientific) was covalently fluorescently labeled with Mirus Label IT Cy3 (MoBiTec GmbH) according to the manufacturer's protocol using a 1:1 (volume:weight) ratio of Label IT reagent to nucleic acid[55]. *H. pylori* liquid cultures (BB-FBS) were grown under microaerobic conditions to $OD_{600}$ of 0.8–1.0 and split into two sub-cultures of 2 ml each. The cultures were separately incubated under microaerobic and aerobic conditions at 37 °C for 1 h. Subsequently, 100 µl of the cells were incubated under microaerobic conditions for 15 minutes with 1 µg/ml of Cy3 λ DNA. Cells were centrifuged (7000 × g, 3 min) and suspended in 30 µl of TSB-FBS supplemented with 2 U of DNase I (Thermo Fisher Scientific). After 5 min incubation at 37 °C, bacteria were diluted five times in TSB-FBS, and 7.5 µl was spread on a 1% low melting agarose pad. Fluorescence microscopy was performed using a Zeiss Axio Imager M1 microscope with a plan 100x/1.3 objective and bright-field. Cy3 was visualised using a mercury light source (HBO100) and a filter set with excitation at 550 nm (bandwidth 25 nm), emission at 605 nm (bandwidth 70 nm) and a dichroic beamsplitter at 570 nm. Regions of interest (ROIs) were set manually and analyzed by Axiovision 4.8. Each experiment was performed using at least three biological replicates.

## Determination of transformation rate

The transformation rate was determined, as previously shown by Kruger et al.[28], with minor modifications. *H. pylori* liquid cultures were grown in BB-FBS under microaerobic conditions to $OD_{600}$ of 0.8–1.0. Cells were harvested by centrifugation (7000 × g, 3 min) and suspended in 100 µl of TSB-FBS (pH 7.5, filtered) to $OD_{600}$ ~ 1. Then, 1 µg/ml of a 1022-bp *rpsL* (A128G) PCR fragment was added to the cell suspension[82]. The mixture was incubated for 1 h at 37 °C under microaerobic or aerobic conditions. Next, cells were centrifuged (7000 × g, 3 min) and suspended in 50 µl of TSB-FBS supplemented with 2 U of DNase I (Thermo Fisher Scientific). After 5 min incubation at 37 °C, bacteria were directly spotted on CBA-B plates and further incubated under microaerobic conditions at 37 °C for 22 ± 2 h. Subsequently, bacteria were harvested in TSB-FBS, and serial dilutions were performed. Cells were spread on CBA-B, and CBA-B plates supplemented with 20 µg/mL streptomycin to count the number of all cells used in transformation and streptomycin-resistant mutants,

respectively. The colony forming units (CFU) of each group were measured, and the transformation rate was determined. Each experiment was performed using three biological replicates.

## Glucose uptake assay

Glucose uptake by *H. pylori* culture during growth was measured using glucometer Countour Plus ELITE and Contour Plus strips. Glucometer allowed for fast and reliable glucose consumption measurements by *H. pylori* (Fig. S11d). 12 ml of the TSBΔD-FBS medium supplemented with 100 mg/dl glucose were inoculated with the *H. pylori* N6 wild-type and mutant strains to $OD_{600} = 0.025$ or with the *H. pylori* P12 wild-type and mutant strains to $OD_{600} = 0.015$ and grown under microaerobic conditions. $OD_{600}$ of each culture was periodically measured until the *H. pylori* microaerobic culture reached the late stationary phase. CFU determined the total amount of cells in the medium. At $OD_{600} = 1$, the number of cells equals approx. $2 \times 10^8$/ml for each strain. The glucose consumption per cell was determined by dividing the change in glucose concentration in the growth medium between the inoculated culture (time 0) and the late logarithmic phase time point (time F, approx. 20–25 h of growth) by the number of cells in 1 ml of culture at the late logarithmic phase (time F). Each experiment was performed using three biological replicates.

## ATP assay

The ATP level was measured with the BacTiter-Glo™ Assay. *H. pylori* cultures were grown under microaerobic conditions to $OD_{600}$ of 0.7–1.0 in TSBΔD-FBS medium supplemented with 100 mg/dl glucose. Cells were diluted to $OD_{600} = 0.1$ with fresh medium. Next, 50 μl of bacteria were mixed with 50 μl of BacTiter-Glo™ (Promega) and incubated at room temperature for 5 min. The luminescence was measured with a CLARIOstar® plate reader on opaque-walled multiwell plates (SPL Life Sciences; 2-200203). Each experiment was performed using three biological replicates.

## Protein expression and purification

The recombinant Strep-tagged HP1021 protein was purified according to the Strep-Tactin manufacturer's protocol (IBA Lifesciences). Briefly, *E. coli* BL21 cells (1 l) carrying the pET28/StrepHP1021 vector were grown at 37 °C. At an optical density ($OD_{600}$) of 0.8, protein synthesis was induced with 0.05 mM IPTG for 3 hours at 37 °C. The cultures were harvested by centrifugation (10 min, $5000 \times g$ and 4 °C). The cells were suspended in 20 ml of ice-cold buffer W (100 mM Tris-HCl, pH 8.0; 300 mM NaCl; and 1 mM EDTA) supplemented with cOmplete, EDTA-free protease inhibitor Cocktail (Roche)), disrupted by sonication and centrifuged (30 min, $31,000 \times g$ and 4 °C). The supernatant was applied onto a Strep-Tactin Superflow high-capacity Sepharose column (1 ml bed volume, IBA). The column was washed with buffer W until Bradford tests yielded a negative result and then washed with 5 ml of buffer W without EDTA. The elution was carried out with approx. $6 \times 0.8$ ml of buffer E (100 mM Tris-HCl, pH 8.0; 300 mM NaCl; and 5 mM desthiobiotin). Protein purity was analyzed by SDS-PAGE electrophoresis using a GelDoc XR+ and ImageLab software (BioRad). The fractions were stored at −20 °C in buffer E diluted with glycerol to a final concentration of 50%.

## Electrophoretic mobility shift assay (EMSA)

PCR amplified DNA probes in two steps. DNA fragments were amplified in the first step using unlabeled primers P29-P30 (pcomB8) or P31-P32 (pvacA) and a *H. pylori* N6 genomic DNA template (Supplementary Data 3 and S4). The forward primers were designed with overhangs complementary to P33, FAM–labeled primer. The unlabeled fragment was purified and used as a template in the second round of PCR using the P33 FAM-labeled primer and the reverse primer P30 or P32 used in the first step. The *oriC2* region was amplified by PCR using P33-P34 primer pairs and a specific pori2 template (Supplementary Data 3),

giving the FAM-*oriC2* probe. FAM-labeled DNA (5 nM) was incubated with the HP1021 protein at 37 °C for 20 min in Tris buffer (50 mM Tris–HCl, pH 8.0; 100 mM NaCl and 0.2% Triton X-100). The complexes were separated by electrophoresis on a 4% polyacrylamide gel in 0.5 × TBE (1 × TBE: 89 mM Tris, 89 mM borate and 2 mM EDTA) at 10 V/cm in the cold room (approx. 10 °C). The gels were analyzed by a Typhoon 9500 FLA Imager and ImageQuant software.

## Catalase activity

*H. pylori* P12 liquid cultures at the late logarithmic phase of growth ($OD_{600} = 0.8–1.0$) were collected by centrifugation (5 min, $4000 \times g$) to obtain $OD_{600} = 1$ (approx. $2 \times 10^8$ cells/ml) per sample and resuspended in 1 ml of 1 × PBS. 10 μl of cells' suspension of each strain were simultaneously mixed with 10 μl of 30% $H_2O_2$. The catalase activity was measured by observation of the production of air bubbles and captured by a camera.

## Construction of the *H. pylori* mutant strains

*H. pylori* mutant strains were constructed using a homologous recombination approach as described by Ge and Taylor[83]. Briefly, *H. pylori* cells were plated from the stock to CBA-B plates and incubated under microaerobic conditions for 24 h. Next, cells were plated on CBA-B plates and incubated for approximately 5 h in the form of a micro-lawn of approx. 10 mm in diameter. Then, the micro-lawn was spotted with 10 μg of purified recombinant plasmids and grown under microaerobic conditions. After 24 h of growth, the culture was plated on a selective medium and grown for five days to obtain single colonies of transformants.

**H. pylori P12 ΔHP1021**. The *H. pylori* P12 ΔHP1021 mutant in which the HP1021 gene was deleted from the chromosome was constructed as described previously for the N6 strain using the pTZ57R/TΔHP1021 plasmid[20] The allelic exchange was verified by PCR using the P13-P14 primers homologous to the chromosomal regions external to the designed recombination sites. The lack of HP1021 was verified by Western blot analysis using a rabbit polyclonal anti-HP1021 antibody[33] (serum dilution 1:2500) and an ECL Anti-Rabbit IgG, HRP-linked whole antibody (GE Healthcare, dilution 1:3000) (Fig. S7c, d).

**H. pylori P12 COM/HP1021**. The *H. pylori* P12 COM/HP1021 mutant with the restored HP1021 gene on the chromosome was constructed as described previously for the N6 strain using the pUC18/HP1021com plasmid[20] The allelic exchange was verified by PCR using the P13-P14 primers homologous to the chromosomal regions external to the designed recombination sites. Additionally, the presence of HP1021 was verified by Western blot analysis using a rabbit polyclonal anti-HP1021 antibody[33] (serum dilution 1:2500) and an ECL Anti-Rabbit IgG, HRP-linked whole antibody (GE Healthcare, dilution 1:3000) (Fig. S7c, d).

**H. pylori N6 ΔgluP**. The *H. pylori gluP* deletion construct (pUC18/ΔgluP) was prepared as follows (Fig. S11c). The upstream and downstream regions of *gluP* were amplified by PCR using the P1-P2 and P5-P6 primer pairs, respectively and *H. pylori* 26695 genomic DNA as a template. The *aphA-3* cassette was amplified using the P3-P4 primer pair; pTZ57R/TΔHP1021[33] was used as a template. The resulting fragments were purified on an agarose gel. Subsequently, the PCR-amplified fragments and the SmaI digested vector pUC18 were ligated according to the method described by Gibson in 2009[84] *E. coli* DH5α competent cells were transformed by heat shock. The DNA fragment cloned in pUC18 was sequenced. Subsequently, *H. pylori* N6 was transformed with the pUC18/ΔgluP plasmid, and the transformants were selected by plating on CBA-B plates supplemented with kanamycin. The allelic exchange was verified by PCR using the P10-P11 primer pair homologous to the chromosomal regions external to the

designed recombination sites. The lack of *gluP* was verified by PCR using the P7-P12 primer pair (Fig. S11e).

**H. pylori N6 COM/gluP.** The *H. pylori gluP* complementation strain was constructed as follows (Fig. S11c). The region upstream of *gluP* and the *gluP* gene was amplified by PCR using the P1-P7 primer pair, while the downstream region flanking *gluP* was amplified by PCR using the P5-P6 primer pair; both regions were amplified using *H. pylori* 26695 genomic DNA as a template. The *cat* cassette was amplified using the P8-P9 primer pair; pUC18/HP1021com[20] was used as a template. The resulting fragments were purified on an agarose gel. Subsequently, the PCR-amplified fragments and the SmaI digested vector pUC18 were ligated according to the method described by Gibson in 2009[84] *E. coli* DH5α competent cells were transformed by heat shock. The cloned insert was sequenced. Subsequently, the obtained plasmid was used in the natural transformation of *H. pylori* N6. The transformants were selected by plating on CBA-B plates supplemented with chloramphenicol. The allelic exchange was verified by PCR using the P10-P11 primers homologous to the chromosomal regions external to the designed recombination sites. Additionally, the presence of *gluP* was verified by PCR using the P7-P12 primer pair (Fig. S11e).

### Statistics and reproducibility

Statistical analysis was performed using GraphPad Prism (version 8). All in vivo experiments were repeated at least three times, and data were presented as mean ± SD. The statistical significance between the two conditions was calculated by paired two-tailed Student's t-test. The statistical significance between multiple groups was calculated by one-way ANOVA with Tukey's post hoc test. $P < 0.05$ was considered statistically significant. The EMSA, Western blot, and catalase activity assay experiments were repeated twice with similar results.

### Reporting summary

Further information on research design is available in the Nature Portfolio Reporting Summary linked to this article.

## Data availability

The RNA-seq FASTQ and processed data generated in this study have been deposited in the ArrayExpress database (EMBL-EBI) under accession code E-MTAB-13025 (https://www.ebi.ac.uk/biostudies/arrayexpress/studies/E-MTAB-13025?key=bf2d8677-1b26-4f1e-ac5f-b1c78c7590be). The ChIP-seq FASTQ and BED files generated in this study have been deposited in the ArrayExpress database (EMBL-EBI) under accession code E-MTAB-13026 (https://www.ebi.ac.uk/biostudies/arrayexpress/studies/E-MTAB-13026?key=2b27b413-e118-4802-859e-5da57ff1ead2). The raw proteomics data, MaxQuant search results, and the used protein sequence database generated in this study have been deposited in the ProteomeXchange Consortium via the PRIDE partner repository[85] under accession code PXD041978. *Helicobacter pylori* 26695 reference genome is deposited in the National Center for Biotechnology Information under accession code NC_000915.1 (www.ncbi.nlm.nih.gov/nuccore/NC_000915.1?report=genbank). Source data are provided in this paper.

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

## Acknowledgements

This work has been supported by the OPUS 17, Project Number 2019/33/B/NZ6/01648, founded by the National Science Centre, Poland, to A.Z.-P. This work has been supported by the EPIC-XS, Project Number 823839, funded by the Horizon 2020 Program of the European Union to C.L. We thank Julia Golz for her help in Cy3-λ DNA preparation. We thank Franziska Hackbarth for technical assistance at BayBioMS, as well as Miriam Abele for mass spectrometric support. The open-access publication of this article was funded by the OPUS 17 (2019/33/B/NZ6/01648, National Science Centre, Poland) and statutory funds from the Ludwik Hirszfeld Institute of Immunology and Experimental Therapy, Polish Academy of Sciences.

## Author contributions

M.N. and A.Z.-P. conceived and designed the experiments. K.S. supported the DNA uptake part. M.N. and J.M. performed the experiments. M.N., A.S., R.K., C.M., C.L., and A.Z.-P. analyzed the data. M.N., A.S., R.K., C.M., C.L., and A.Z.-P. wrote the manuscript. M.N., A.S., J.M., R.K., C.M., C.L., K.S., and A.Z.-P. revised the manuscript. C.L. and A.Z.-P. provided funding.

## Competing interests

The authors declare no competing interests.
