## [Peer Review File · Nature Communications]

REVIEWER COMMENTS

Reviewer #1 (Remarks to the Author):

Line 103; The “S” of “WTS” and “HP1021S”...I understand it means aerobic conditions, but (and this is a minor point, I know) but why? This is not a standard denotation, as far as I know.

A comment regarding “oxidative stress”: While I agree 21% O₂ vs 5% O₂ is an oxidative stress, I question the physiologic significance of such an artificial oxidative stress. It would seem to me that an oxidative stress reflective of that mounted by host inflammatory response, comprised of ROS created by host cells would be the better stressor. I don't think this calls any of the authors results into question, it just makes me question how relevant these gene regulatory events are to the actual infection of a human host.

Line 113: “The same binding sites”: confusing. Does this mean the 100 binding sites OR katA, gluP, etc.? The wording here is confusing. Of course the same binding “sites” were present in WT and WTS. I think you're stating that HP1021 was bound to these sites under both aerobic AND microaerobic conditions. Needs a bit of clarity.

119: EMSA and ChIP-qPCR need to be inverted to match the order of the data in Figure S2

120-121: The authors contend that aerobic conditions result in the dissociation of HP1021 from pVacA. However it is unclear as Figure S2B appears to compare log₂FC of reads of “HP1021-WT under microaerobic conditions to WT under aerobic conditions. It's not quite clear to me that this is the appropriate comparison to allow the stated conclusion. If this is the appropriate comparison, then the authors need to point out and explain the log₂FC of reads in the genes of the upstream operon are increased as well. Need clarity here.

Lines 126-157. This is extremely dense and takes VERY careful reading and could use a bit of clarity, the Lines 158-162 sum it up well!

The section on proteomic changes is also highly complex and difficult to read, however, the authors do a solid job explaining the potential cause of the relative absence of proteomic changes in the face of oxidative stress; failure to synthesize proteins during the stress. Still, this section bears some simplification and shortening.

The authors mention the role of HP1021 in response to RNS, yet fail to put into context the role of the TCS, CrdRS or even cite the 2015 Mol. Micro paper by Hung et al. This needs to be considered to make the context more complete.

Overall, a very extensive analysis and well written, even if a bit difficult to read and lengthy. It would benefit from a significant shortening and table or tables in the main paper listing genes and proteins affected, rather than in spreadsheets only form in the supplemental.

Reviewer #2 (Remarks to the Author):

The study presented in this manuscript provides a comprehensive description of the regulon controlled by the HP1021 atypical response regulator of the human pathogen *Helicobacter pylori*.

This regulator, which seems to be able to work without the need of phosphorylation, has been previously shown to be involved in the replication of *Helicobacter* chromosome. More recently, it has been demonstrated that HP1021 works as a redox switch and controls the transcription of some genes in response to this kind of signal. In this work, a multi-omics approach (combining RNA-seq – proteomics – Chromatin immunoprecipitation-sequencing) has been set-up and adopted to provide a comprehensive picture of the regulatory role of HP1021 in *Helicobacter*. These data complement recent “biochemical” results about HP1021 redox-response and significantly enrich the understanding that the scientific community has on transcriptional regulation in this important human pathogen. After presenting the major findings deriving from “omics” techniques, in the second part of the results and discussion sections, the authors focus on and deepen some cellular pathways and processes. Of note, in my opinion, is the demonstration of the role of HP1021 in the regulation of the first step involved in DNA uptake from the external environment (through the regulation of promoters controlling comB genes-containing operons). This latter part of the manuscript (DNA uptake assays, but also glucose consumption and measurement of ATP level) adds great value to the whole story.

Starting from the beginning of the manuscript, abstract and introduction are clear and straightforward.

Results and discussion section: I appreciate the decision to merge the presentation of the results with the discussion part; this makes the reading smoother and less scattered. Throughout the manuscript, data are well presented, in general, it is well written and figures are informative and clear.

The approaches used in this study are sound. The quality of the data is good and the use of statistics is appropriate. The methodologies used are sound and very well described, with details to reproduce the work.

References are appropriate to credit previous works.

I have just a few comments and issues:

- ChIP-seq data analysis: the authors associate binding sites with a promoter when they are located between positions -250 to +250 relative to a TSS. In my opinion, this is a very wide range, considering

the compactness of the bacterial promoters and the average size of the 5'-UTR regions in *Helicobacter* (50% of the 5'UTRs are 20–40 nucleotides in length - <https://doi.org/10.1038/nature08756>). That analysis may have led to an overestimation of promoter-associated HP1021 binding sites (84 out of 100 according to this analysis) at the expense of intracistronic binding sites, which are not negligible for other *Helicobacter* (such as Fur) and non-*Helicobacter* (*E. coli* CRP) regulators.

This could also explain the fact that only a minority of the HP1021 binding sites identified by ChIP are associated with regulation (lines 160-163). What was the rationale behind the choice of this parameter?

- The sentence "The same binding sites were detected in *H. pylori* WT and WTS cells, which suggests that HP1021-DNA interaction mechanisms that control gene transcription may include remodelling of the complex upon oxidative stress rather than protein dissociation from its binding sites" (lines 113-116) should be clarified and rewritten more explicitly, I am not sure the reader can grasp the meaning.

- Also the sentence "To conclude, the HP1021 regulon includes 498 genes (HP1021 not included), out of which 407 respond to oxidative stress, HP1021 and other additional regulators possibly control the response of 4 genes and 87 respond to unknown conditions Fig. 1a)" (lines 158-160) is not clear and, in my opinion, should be rewritten.

- I am not convinced that it is correct to assign as part of the HP1021 regulon genes that respond transcriptionally to oxidative stress, but do not vary in the absence of HP1021. This is what I understood from the text "The significant number of genes assigned to the HP1021 regulon suggests that HP1021 regulates many pathways and processes. Indeed, the performed functional analysis (eggNOG) indicated that the expression of genes belonging to many Clusters of Orthologous Groups (COG) was affected by oxidative stress or by the lack of HP1021 at the transcription and/or translation steps" (lines 193-196). I see them as genes that respond to oxidative stress, in a regulon-independent manner. Please comment and clarify.

We thank the Reviewers for their comments. We answered all questions and made the required corrections in the manuscript (a file submitted in a track changes mode). Additional changes made in the manuscript are listed below the answers to Reviewers.

REVIEWER COMMENTS

Reviewer #1 (Remarks to the Author):

Line 103; The “S” of “WTS” and Δ HP1021S”. I understand it means aerobic conditions, but (and this is a minor point, I know) but why? This is not a standard denotation, as far as I know.

Aerobic conditions cause oxidative stress for *H. pylori*, so to distinguish between non-stressed and stressed cells, we added the letter “S” (abbreviation of “stress”). To make this more apparent, we noted that aerobic conditions cause stress (lines 103-105).

A comment regarding “oxidative stress”: While I agree 21% O₂ vs 5% O₂ is an oxidative stress, I question the physiologic significance of such an artificial oxidative stress. It would seem to me that an oxidative stress reflective of that mounted by host inflammatory response, comprised of ROS created by host cells would be the better stressor. I don’t think this calls any of the authors results into question, it just makes me question how relevant these gene regulatory events are to the actual infection of a human host.

According to literature, O₂ diffuses through membranes, acquires electrons from the reduced cofactors of flavoproteins and finally turns into superoxide (O₂⁻) and hydrogen peroxide (H₂O₂) (doi.org/10.1186/s12934-014-0181-5, doi.org/10.1186/s12934-014-0181-5). We have previously shown that: 1/ *H. pylori* cells lacking HP1021 were more sensitive to O₂⁻ (triggered by paraquat) and H₂O₂ than the wild-type strain, and 2/ aerobic stress modified cysteine residues of HP1021 *in vivo* and that *H. pylori* mutant producing cysteine-less HP1021 did not respond to oxidative stress as WT did. Taking into account that superoxide (O₂⁻) and hydrogen peroxide (H₂O₂) are oxygen species produced by human immune cells and epithelial cells in response to *H. pylori* infection (doi.org/10.1016/j.freeradbiomed.2016.09.024), we assumed that by using 21% O₂ we would trigger the formation of similar oxygen species that immune cells produce.

Line 113: “The same binding sites”: confusing. Does this mean the 100 binding sites OR katA, gluP, etc.? The wording here is confusing. Of course the same binding “sites” were present in WT and WTS. I think you’re stating that HP1021 was bound to these sites under both aerobic AND microaerobic conditions. Needs a bit of clarity.

We rewrote the sentence and the paragraph (lines 116-125), and we think it is less confusing now.

119: EMSA and ChIP-qPCR need to be inverted to match the order of the data in Figure S2

It was corrected in the text, and now the order in the text and figures match.

120-121: The authors contend that aerobic conditions result in the dissociation of HP1021 from pVacA. However it is unclear as Figure S2B appears to compare log₂FC of reads of Δ HP1021-WT under microaerobic conditions to WT under aerobic conditions. It’s not quite clear to me that this is the appropriate comparison to allow the stated conclusion. If this is the appropriate comparison, then the authors need to point out and explain the log₂FC of reads in the genes of the upstream operon are increased as well. Need clarity here.

Dissociation of HP1021 from *vacA* promoter was shown by ChIP-seq (Fig. S2a) and confirmed by ChIP-qPCR (Fig. S2d). In ChIP-seq, the number of reads detected at the *vacA* promoter region in wild-type

cells under microaerobic conditions (WT, middle panel, grey) was higher than in wild-type cells under aerobic stress (WTS, lower panel, yellow). In ChIP-qPCR, the fold enrichment of the *pvacA* promoter is lower in WTS cells than in WT cells. Fig. S2b presents RNA-seq results and compares *vacA* transcription of WTS and Δ HP1021 to the WT cells. We rewrote this part to better present the results (lines 126-131).

Lines 126-157. This is extremely dense and takes VERY careful reading and could use a bit of clarity, the Lines 158-162 sum it up well!

We rewrote and shortened this paragraph significantly (lines 139-188). We added a Venn diagram to present the general numbers concerning HP1021 regulon better (Fig. 1a, inset). We removed some information (e.g., a few names of the genes which distracted the reader without bringing up too much information), moved information about numbers of genes up- or downregulated at different conditions from the main text to the description of Figure 1 a and c (upper corners of Fig. 1a and 1c, legend). We also reorganized the flow of information to lead the reader through the data better. We hope this paragraph is easier to read and follows the main message concerning transcription regulated by HP1021.

The section on proteomic changes is also highly complex and difficult to read, however, the authors do a solid job explaining the potential cause of the relative absence of proteomic changes in the face of oxidative stress; failure to synthesize proteins during the stress. Still, this section bears some simplification and shortening.

This concise paragraph gives only the most essential details concerning HP1021-dependent proteome changes, combining transcriptome and translome changes and discussing possible discrepancies between these two sets of results. We made some shortening, but, in our opinion, the rest should be kept to give the readers important information concerning proteomic results.

The authors mention the role of HP1021 in response to RNS, yet fail to put into context the role of the TCS, CrdRS or even cite the 2015 Mol. Micro paper by Hung et al. This needs to be considered to make the context more complete.

We added a sentence informing that *H. pylori* uses the CdRS TCS system to respond to RNS stress (lines 210-211). However, we did not go into detail because it would require expanding the sub-thread of the RNS response, which, as recent reports indicate, is still not fully explained. We cited relevant works (Hung et al., 2015, and recent work by Allen et al., 2023), so anyone interested can explore the topic.

Overall, a very extensive analysis and well written, even if a bit difficult to read and lengthy. It would benefit from a significant shortening and table or tables in the main paper listing genes and proteins affected, rather than in spreadsheets only form in the supplemental.

We are afraid that putting a list of nearly 500 genes in the main text is impossible for editorial reasons. The main text includes relevant volcano plots dedicated to presenting large data sets. Some most significantly affected genes are marked on these plots (the name or the number of the gene is depicted), while the rest of the genes are only indicated as color-coded data. The comprehensive supplementary tables are available and present detailed information required for analysis. We tried to reduce the length of our work by shortening the RNA-seq and proteome parts, adding a Venn diagram and additional information in Fig. 1a,c to present data rather than to describe them. We believe that now the manuscript is easier to read.

Reviewer #2 (Remarks to the Author):

The study presented in this manuscript provides a comprehensive description of the regulon controlled by the HP1021 atypical response regulator of the human pathogen *Helicobacter pylori*. This regulator, which seems to be able to work without the need of phosphorylation, has been previously shown to be involved in the replication of *Helicobacter* chromosome. More recently, it has been demonstrated that HP1021 works as a redox switch and controls the transcription of some genes in response to this kind of signal. In this work, a multi-omics approach (combining RNA-seq – proteomics – Chromatin immunoprecipitation-sequencing) has been set-up and adopted to provide a comprehensive picture of the regulatory role of HP1021 in *Helicobacter*. These data complement recent “biochemical” results about HP1021 redox-response and significantly enrich the understanding that the scientific community has on transcriptional regulation in this important human pathogen. After presenting the major findings deriving from “omics” techniques, in the second part of the results and discussion sections, the authors focus on and deepen some cellular pathways and processes. Of note, in my opinion, is the demonstration of the role of HP1021 in the regulation of the first step involved in DNA uptake from the external environment (through the regulation of promoters controlling *comB* genes-containing operons). This latter part of the manuscript (DNA uptake assays, but also glucose consumption and measurement of ATP level) adds great value to the whole story. Starting from the beginning of the manuscript, abstract and introduction are clear and straightforward. Results and discussion section: I appreciate the decision to merge the presentation of the results with the discussion part; this makes the reading smoother and less scattered. Throughout the manuscript, data are well presented, in general, it is well written and figures are informative and clear. The approaches used in this study are sound. The quality of the data is good and the use of statistics is appropriate. The methodologies used are sound and very well described, with details to reproduce the work. References are appropriate to credit previous works. I have just a few comments and issues:

- ChIP-seq data analysis: the authors associate binding sites with a promoter when they are located between positions -250 to +250 relative to a TSS. In my opinion, this is a very wide range, considering the compactness of the bacterial promoters and the average size of the 5'-UTR regions in *Helicobacter* (50% of the 5'UTRs are 20–40 nucleotides in length - <https://doi.org/10.1038/nature08756>). That analysis may have led to an overestimation of promoter-associated HP1021 binding sites (84 out of 100 according to this analysis) at the expense of intracistronic binding sites, which are not negligible for other *Helicobacter* (such as Fur) and non-*Helicobacter* (*E. coli* CRP) regulators. This could also explain the fact that only a minority of the HP1021 binding sites identified by ChIP are associated with regulation (lines 160-163). What was the rationale behind the choice of this parameter?

We set up the parameter to +/-250 bp from TSS start not to miss too many of the possible regulatory sites. mRNA starts at TSS; however, the RNAP polymerase binds upstream of the TSS site, while the transcription factors (TF) may bind beyond the RNAP binding site, either downstream or upstream (nicely depicted in comprehensive studies on *E. coli* TF proteins, doi.org/10.7554/eLife.55308). Thus, the promoter regions with TF binding sites are extended to ca 100-120 bp, possibly longer or shorter depending on the gene. On the other hand, the ChIP-seq algorithms (in our case – edgeR and/or MACS3) identify chromosomal regions bound by TF with a certain precision. In our case, the peaks defining HP1021 binding sites spanned over 100 – 5500 bp (median width of the peak 463 bps, using two algorithms – edgeR and MACS3). Thus, in the algorithm which correlates ChIP-seq main peaks with TSS, the user defines a specific window allowing the ChIP-seq peak sequence to overlap with TSS. For example, in *E. coli* or *Synechocystis sp.*, in which the length of 5'-UTR was similar to *H. pylori* (majority *E. coli* UTRs were 25-30 bp, doi.org/10.1371/journal.pgen.1002867; median *Synechocystis sp.* UTR was 53 bp, [doi: 10.1128/mSystems.00943-21](https://doi.org/10.1128/mSystems.00943-21)), +/- 250 bp or +/-300 bp window, respectively, was used in annotating the peaks to TSS (doi.org/10.1128/jb.00411-18, [doi:10.21769/BioProtoc.2895](https://doi.org/10.21769/BioProtoc.2895)). In our case, ca 50% of ChIP-seq binding sites could be correlated with TSS using a +/-150 bp window (51 sites out of 100 detected). However, in that case, some binding sites would be lost, including *comB2*, *gluP* and *vacA* promoter regions, which transcription changed in HP1021-dependent manner (Figs: 2, 3 and

S2); in addition, the binding of HP1021 to *gluP* and *vacA* was confirmed in ChIP-qPCR and EMSA (Figs. 3 and S2). Thus, considering the limits of ChIP-seq analysis and the literature data, we decided to use +/-250 bp window in our studies.

- The sentence "The same binding sites were detected in *H. pylori* WT and WTS cells, which suggests that HP1021-DNA interaction mechanisms that control gene transcription may include remodelling of the complex upon oxidative stress rather than protein dissociation from its binding sites" (lines 113-116) should be clarified and rewritten more explicitly, I am not sure the reader can grasp the meaning.

We rewrote the sentence and added additional information to clarify it (lines 126-131 and 139-188).

- Also the sentence "To conclude, the HP1021 regulon includes 498 genes (HP1021 not included), out of which 407 respond to oxidative stress, HP1021 and other additional regulators possibly control the response of 4 genes and 87 respond to unknown conditions Fig. 1a)" (lines 158-160) is not clear and, in my opinion, should be rewritten.

We rewrote the sentence, and we hope it is easier to follow. Please note that we rewrote and shortened this paragraph significantly to respond to the comments of Reviewer #1. See also additional information below.

- I am not convinced that it is correct to assign as part of the HP1021 regulon genes that respond transcriptionally to oxidative stress, but do not vary in the absence of HP1021. This is what I understood from the text "The significant number of genes assigned to the HP1021 regulon suggests that HP1021 regulates many pathways and processes. Indeed, the performed functional analysis (eggNOG) indicated that the expression of genes belonging to many Clusters of Orthologous Groups (COG) was affected by oxidative stress or by the lack of HP1021 at the transcription and/or translation steps" (lines 193-196). I see them as genes that respond to oxidative stress, in a regulon-independent manner. Please comment and clarify.

To determine the genes regulated by HP1021 (directly or indirectly), we compared the expression of genes in WT and Δ HP1021 strains under microaerobic conditions. To identify genes connected with oxidative stress response in WT strain (i.e., those genes which respond to stress in WT) and those which are oxidative stress independent (i.e., do not respond to oxidative stress in WT) and to define which of them are controlled by the HP1021 protein we compared the expression of genes in WT and Δ HP1021 strains under aerobic stress.

We found 190 genes differentially expressed under microaerobic conditions in the Δ HP1021 strain compared to the WT strain. The transcription of 103 of 190 genes was also changed in the WTS cells under stress compared to WT. However, only 3 of 103 also changed in Δ HP1021S under oxidative stress compared to Δ HP1021 but not in Δ HP1021S compared to Δ HP1021, while 100 did not respond to oxidative stress in Δ HP1021S cells. Thus, 100 of 190 genes comprise the regulon responding to oxidative stress (see inset Venn diagram in Fig. 1a).

We also found 307 genes whose transcription changed under oxidative stress in WTS cells but under microaerobic conditions expressed in Δ HP1021 as in WT. These genes did not respond to stress in Δ HP1021 (i.e., the transcriptional control over these genes is lost in Δ HP1021); thus, they belong to the HP1021 regulon related to oxidative stress.

87 genes differentially expressed under microaerobic conditions in the Δ HP1021 strain compared to the WT strain are unrelated to oxidative stress because their transcriptions do not change between WT and WTS cells.

Altogether the oxidative stress-dependent regulon comprises 407 genes (100+307), while the entire regulon consists of 497 genes (407+87, plus 3 genes whose transcription is controlled by HP1021 under microaerobic conditions but possibly by additional factors under oxidative stress).

We think that the sentence of the manuscript, cited by the Reviewer, might have been misleading, and we changed it to point out that in each case, the lack of HP1021 dysregulated expression, but the dysregulated genes belonged to two classes: involved or not involved in the cell's response to oxidative stress. We also added a Venn diagram in Fig. 1a (inlet), which we think will also help follow our data.

ADDITIONAL CHANGES MADE IN THE MANUSCRIPT:

1. The number of all genes in the HP1021 regulon changed from 498 to 497. We removed one gene (frpB1) from the list because its transcription was unchanged in Δ HP1021, while it was activated in WTS and Δ HP1021S cells under oxidative stress. The activation of frpB1 under oxidative stress significantly differed in WT and Δ HP1021 strains suggesting that HP1021 helps to activate frpB1 upon stress (WTS-WT fold change of 3.07, Δ HP1021S- Δ HP1021 fold change of 1.01). Nonetheless, we decided to remove frpB1 from the list of genes whose transcription is regulated by HP1021.

2. The number of oxidative stress response genes in the HP1021 regulon changed from 411 to 407. The transcription of 411 genes changed in WTS compared to WT cells (i.e. they responded to oxidative stress). However, the transcription of 3 of them changed also under oxidative stress in Δ HP1021S cells. Thus, these genes, being controlled by the HP1021 protein under microaerobic conditions (Δ HP1021-WT fold changes: HP0546a FC of 1.08, serB (HP0652) FC of -2.9, pfr (HP0653) FC of -3.21), possibly also require other (or additional) factors to control their transcription under oxidative stress. In other words, they belong to the HP1021 regulon, but their regulation under oxidative stress might require other/additional factors. The reason for removing the 4th gene is described in point 1.

It should be noted that these 4 genes were treated as a separate set of genes in the former version of the manuscript, but we think that presenting data in the Venn diagram and subtracting these genes from oxidative stress genes and/or HP1021 regulon makes our data more concise.

3. We changed Figures 1a and c, as described in the answer to the comments of Reviewer #1

4. We noticed some errors in Table S3 in the columns with proteomic data. In some data sheets (e.g., DNA uptake), we did not distinguish proteins that were not significantly changed (NA) from not detected proteins (ND). It is important for data interpretation; thus, we carefully verified and corrected the entire Table S3 so that the data in Table S3 correspond with the data in Table S2.

5. We added information about the number of proteins whose level changed in the abstract, and we rewrote two sentences to distinguish information about genes and proteins differently expressed due to the lack of the HP1021 regulator. We think that this is important to give precise information about how transcriptome and proteome data correlate. We rewrote the abstract to meet the word limit count (150 words max).

6. We have made editorial changes to meet editorial requirements. We changed the titles of the headers (they cannot be longer than 60 characters). We shortened the legend to Fig. 2 (only 350 words are allowed) by removing information not essential for reading the figure. We also changed the font size and type in the headers and subheaders to meet editorial criteria, but we did not mark these changes in the manuscript as it does not affect the manuscript data.

REVIEWERS' COMMENTS

Reviewer #1 (Remarks to the Author):

I believe the authors have addressed the issues appropriately and I feel it is acceptable in the present state.

Reviewer #2 (Remarks to the Author):

The authors have responded convincingly to the comments made in the first round of revisions. In my opinion, the extensive revisions of the text have clarified several points that might have been misleading or difficult to interpret in the earlier version.

With all this in mind and given the importance of the work to the field, I recommend publication of the manuscript in Nature Communications.